# Breast cancer quantitative proteome and proteogenomic landscape

Henrik J. Johansson ⬤ et al.[#]

In the preceding decades, molecular characterization has revolutionized breast cancer (BC) research and therapeutic approaches. Presented herein, an unbiased analysis of breast tumor proteomes, inclusive of 9995 proteins quantified across all tumors, for the first time recapitulates BC subtypes. Additionally, poor-prognosis basal-like and luminal B tumors are further subdivided by immune component infiltration, suggesting the current classification is incomplete. Proteome-based networks distinguish functional protein modules for breast tumor groups, with co-expression of EGFR and MET marking ductal carcinoma in situ regions of normal-like tumors and lending to a more accurate classification of this poorly defined subtype. Genes included within prognostic mRNA panels have significantly higher than average mRNA-protein correlations, and gene copy number alterations are dampened at the protein-level; underscoring the value of proteome quantification for prognostication and phenotypic classification. Furthermore, protein products mapping to non-coding genomic regions are identified; highlighting a potential new class of tumor-specific immunotherapeutic targets.

---

[#]A full list of authors and their affiliations appears at the end of the paper.

Research efforts over the preceding decades have led to immense progress in our understanding of the molecular heterogeneity of tumors originating in the same tissue, solidifying a long-proposed idea that single effective organ-of-origin specific treatments are not adequate. This realization fostered the need for in-depth molecular characterization to stratify patients into treatment courses that target their individually unique tumors. This principle was first applied to breast cancer when Botstein et al. classified 42 tumors into molecular subtypes based on their mRNA signatures[1]. These original classifications have proven extremely robust and are still widely used to predict prognosis and design therapeutic regimens[2,3].

To aid in clinical implementation, a set of 50 transcripts (collectively known as PAM50) were established for the five subtypes (basal-like, HER2, luminal A & B, and normal-like) and surrogate immunohistochemistry (IHC) markers (ER, PR, HER2, and Ki67) were implemented to partially recapitulate the stratifying and prognostic information garnered in the original studies. However, multigene expression assays (e.g., MammaPrint[TM], Oncotype DX[TM], and Prosigna ROR[TM]) are not readily available to all patients, and despite progress in the development of pathology-based surrogate PAM50 markers, one out of three patients are still potentially misclassified[2,3].

Parallel advancements in high-throughput protein quantification techniques have enabled the burgeoning of protein-based molecular characterization of breast tumors. In theory, these classifications are a more accurate reflection of functional heterogeneity and stronger predictors of therapeutic response, as cellular function and pharmaceutical intervention are largely mediated at the protein level. Though mRNA-based classifications have had great clinical utility, certain shortcomings may be attributable to varying protein–mRNA abundance correlations[4,5] and the inability of mRNA measurements to capture ligand-mediated interplay between tumor and host and characterize the extracellular space.

The immaturity of the field of high-throughput proteomics relative to transcriptomics is a major obstacle for protein-based studies to drastically alter the clinical approach to breast cancer, as Botstein et al. did nearly two decades ago. However, recent breakthroughs have offered a glimpse of that potential. High-throughput mass spectrometry-based protein quantification of PAM50 gene products was found to partially recapitulate the patient stratification offered by the original mRNA-based PAM50 subtypes[5,6] and unbiased analysis of protein expression signatures has identified a subset of tumors, not identified by mRNA analysis, as being associated with a high degree of tumor differentiation and improved patient outcome[5,7–9].

The continued advances in proteomic and genomic technologies have led to the emergence of the field of proteogenomics. Proteogenomics studies link aberrations observed at the protein level to genomic events or properties, such as mutations, insertions/deletions, substitutions, and single-nucleotide polymorphisms[4,5,10,11]. These analyses can identify protein products of genomic regions, previously thought to be silent and/or specifically expressed in transformed tissue that could represent neoantigens[11,12]. The discovery of neoantigens has potential to be an extremely powerful tool in the design of immunotherapies.

Herein, we present an in-depth quantitative profile of the proteomes of 45 breast tumors, 9 represented from each of the 5 PAM50-based molecular classifications. We demonstrate a remarkable quality of relative quantification by examining protein complex member correlations across tumor samples and are the first to recapitulate the current mRNA-based molecular classifications with an unsupervised analysis of whole-proteome data. We then use the high-quality proteome profiles as a base to interpret multiple layers of systems measurements collected on the same tumors, including those of mRNA expression, genome copy-number alterations, single-nucleotide polymorphisms, phosphoprotein levels, and metabolite abundances. Independent layers of analyses reveal novel immunohistochemical biomarker candidates to more reliably stratify difficult-to-classify patients for treatment options, provide a proteome-based framework to assess prognosis for those straddling treatment class assignments, link immune cell infiltration and tumor extracellular matrix composition to prognosis, and connect molecular classification to metabolic phenotype. Furthermore, the depth and quality of proteome profiling enables application of proteogenomic analyses and the discovery of neoantigens arising from tumor-specific variants of known proteins and regions of the genome previously thought to be noncoding. Finally, the comprehensive data collected in this study are presented as an online resource for the breast cancer research community to explore and to test new hypotheses within their areas of expertise (www.breastcancerlandscape.org).

## Results

**MS-based proteomics quantification of a breast tumor cohort.** Nine patients classified into each of the five PAM50 subtype groups were selected from the Oslo2 study cohort to ensure tumor diversity is represented (denoted Oslo2 Landscape cohort) (Fig. 1)[13,14]. LC-MS/MS-based protein quantification was performed as described in the Supplementary Methods section[11,12].

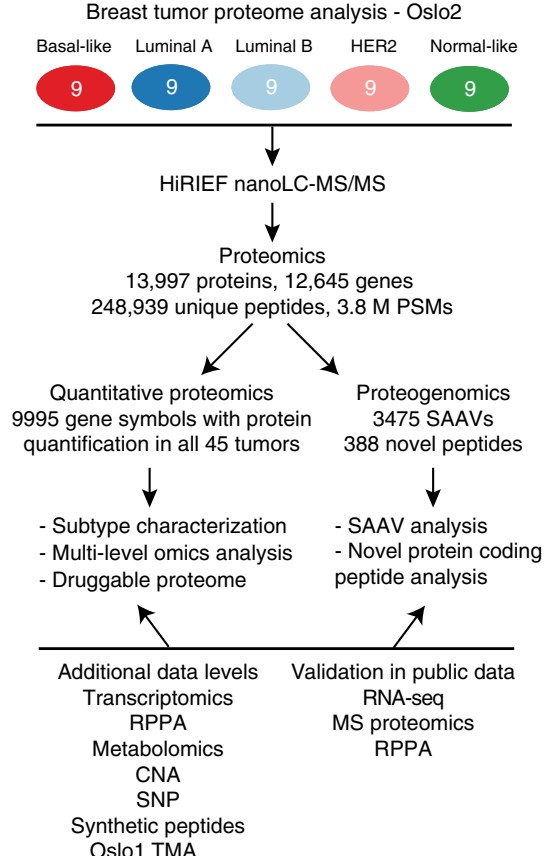

**Fig. 1** Proteomics workflow overview. Quantitative proteome and proteogenomics analyses, and additional data levels used for validation and multi-level omics analysis. PSM peptide spectrum match, SAAV single amino acid variant, HiRIEF high-resolution isoelectric focusing, RPPA reverse phase protein array, CNA copy number alteration, SNP single-nucleotide polymorphism, TMA tissue microarray

In all, 13,997 protein products of 12,645 genes were identified at a 1% protein false-discovery rate (FDR) based on 248,949 identified unique peptides (Fig. 1, Supplementary Fig. 1A, B, Supplementary Data S1). The subset of 9995 proteins quantified (with a median of 12 unique peptides/protein and 24 PSMs/protein for quantification) in each of the 45 tumors, based on gene symbol centric quantification (denoted proteins henceforth), is used for all quantitative proteome analyses (i.e., the quantified proteome) (Supplementary Fig. 1C–H).

Robustness of protein identification/quantification was examined by searching raw MS spectra using parallel methods (MS-GF + Percolator[15,16] and Andromeda in MaxQuant) and performing reverse phase protein lysate assays (RPPA) on sections of the same tumors. Both spectral search methods yield similar protein identifications (Supplementary Fig. 1I), 60% of whose quantities are positively correlated with RPPA findings (Supplementary Fig. 1J)[13,17], and MS-based profiles of BC hallmark proteins are consistent with well-established characteristics of tumor PAM50 classifications (Supplementary Fig. 1K).

**Correlation analysis of tumor proteomes and metabolomes**. Unsupervised hierarchical clustering of proteome profiles stratifies tumors largely in agreement with the PAM50 subtypes (Fig. 2a, Supplementary Fig. 2, Table S1). Basal-like, normal-like, and luminal A groups are distinguished; however, the luminal B and HER2 subtypes are intermixed, indicating similarities in the molecular phenotype. The validity of these mixed classifications is further supported by tumor-transcript profiles of both PAM50 subtypes correlating with either subtype centroid (Supplementary Fig. 2A) and by clinically HER2+ patients often receiving a conflicting mRNA-based classification[18].

Analogous clustering of relative protein quantities, across tumor samples, groups proteins in accordance with their known co-functions in BC biology. Gene ontology enrichment analysis reveals that proteins considered luminal markers, basal markers, or members of the HER2 amplicon, localized to the mitochondria or Golgi apparatus, related to proliferation, transcription, adipose tissue, erythrocytes, immune response, or the extracellular matrix are closely correlated and coregulated with members of their respective groups. Of note, plasma and erythrocyte proteins originate outside of the tumor and would not be detected by transcriptional profiling, demonstrating the unique capability of proteome profiling to consider the tumor in the context of systemic functions of the host (Fig. 2a, Supplementary Fig. 2). In addition, tumor composition correlates between MS- and histopathology-based evaluation (Supplementary Fig. 3).

RPPA was performed with phosphorylation-specific antibodies against 41 known cancer-related regulators of cell signaling to explore their impact on the quantified tumor proteome. Hierarchical clustering of phosphoprotein correlation profiles (RPPA-quantified phosphoprotein abundance to MS-quantified protein abundance) divides phosphoproteins into four distinct groups (Supplementary Fig. 2F, I). Phosphorylation of proteins of group 1 (including CHEK1, CDKN1B, and MAP2K1), group 2 (including tyrosine kinases MET, EGFR, and ERBB3), group 3 (including ERBB2, EGFR, and downstream targets JUN and SC1), and group 4 (including ESR1 and RPS6KA1), respectively, regulate proteins associated with proliferation, blood plasma, the *HER2* amplicon, and the luminal subtype. Interestingly, HGF and EGFL7 (MET and EGFR ligands) and HGFAC (activates HGF) are in the MS-based protein correlation group associated with blood plasma, indicating a possible pathway of activation through phosphorylation of proteins of group 2.

PAM50 subtype assignments are based on mRNA profile distance to subtype centroid as defined by Parker et al.[19]. High PAM50 subtype agreement with correlation-based hierarchical clustering of tumor protein expression profiles considering only the 37 PAM50 gene members in the quantified proteome demonstrates the patient-stratifying information contained within the entire proteome is derived from a smaller subset (Fig. 2b). Centroid-based subtype assignments are validated by hierarchical clustering of transcript measurements from the same 37, and all 50, PAM50 genes (Supplementary Fig. 4A, B); though unsupervised hierarchical clustering of correlations to each PAM50 subtype centroid demonstrates some ambiguity in the classification (Supplementary Fig. 2D).

Core sets of tumors whose proteomes are representative of a proteome-based grouping are defined using unsupervised clustering based on high-variance protein ($n = 1334$) abundance profiles (Supplementary Methods), producing six consensus core tumor clusters (CoTC) (Fig. 2c, Supplementary Fig. 4C–K). CoTC assignments overlap with PAM50 normal-like and luminal A classifications, but divide PAM50 basal-like tumors into two groups, and combine HER2 and luminal B while maintaining a separate group of luminal B PAM50 subtype tumors (Fig. 2c, d, Supplementary Fig. 3I). Unsupervised clustering of CPTAC breast tumor proteomes[5], using the overlapping high-variance proteins (632 of 1334), identifies three tumor clusters that resemble CoTC1 (basal-like), CoTC3 (luminal A), and CoTC6 (a mix of luminal B and HER2) (Supplementary Fig. 5A, B). Of note, the CPTAC patient cohort does not have a defined normal-like tumor subtype.

The CoTC groups, composed of PAM50-classified basal-like tumors (CoTC1 and 2), are distinguished by differential expression of immune markers, E2F and MYC targets, along with G2M checkpoint-related proteins (Fig. 2e). The luminal and HER2 dominated CoTC groups (3, 4, and 6) are stratified by differential enrichment for proteins related to the estrogen response, E2F targets, G2M checkpoint proteins, and MYC targets (Fig. 2e). Tumors with similar immune enrichment as those of the CoTC2 and CoTC4 groups containing two tumors each are observed in the whole Oslo2 cohort (Supplementary Fig. 5C–E). Pairwise inspection reveals proliferation-related and interferon and estrogen response proteins account for the largest share of variability between proteome-based tumor groups (Supplementary Fig. 6).

Finally, tumors displaying marked glycolytic characteristics, as determined by stratification based on profiled metabolite abundances, are all members of CoTC6 (Fig. 2f). Depleted glucose and elevated lactate/alanine indicate glucose may be rapidly oxidized to pyruvate followed by conversion to lactate or alanine. Furthermore, elevated MKI67 (a marker of proliferation) in these tumors is consistent with the Warburg effect (Fig. 2g)[20]. Overlapping pursuant classifications with those based on an independent measurement of cellular function is emblematic of the value added by proteome-based profiling.

**Proteome characterization reveals tumor subclass processes**. Protein abundances of a number of known complex members are exceedingly correlated, as exemplified by condensin I, MCM, GINS, condensin II, mitotic 14 s cohesin I, and DNA polymerase alpha complexes (Fig. 3a, Supplementary Fig. 7A, B). Although the common biological process involvements of protein complex members is expected to be evidenced by correlation[21], the physical nature of component interactions or their tightly related functions may exacerbate this effect. Indeed, proteins with known interactions, as reported by Biogrid or CORUM, have more correlated abundances. Moreover, these elevated correlations are substantially more distinguished at the protein as opposed to the transcript level (Fig. 3b, Supplementary Fig. 7B, C).

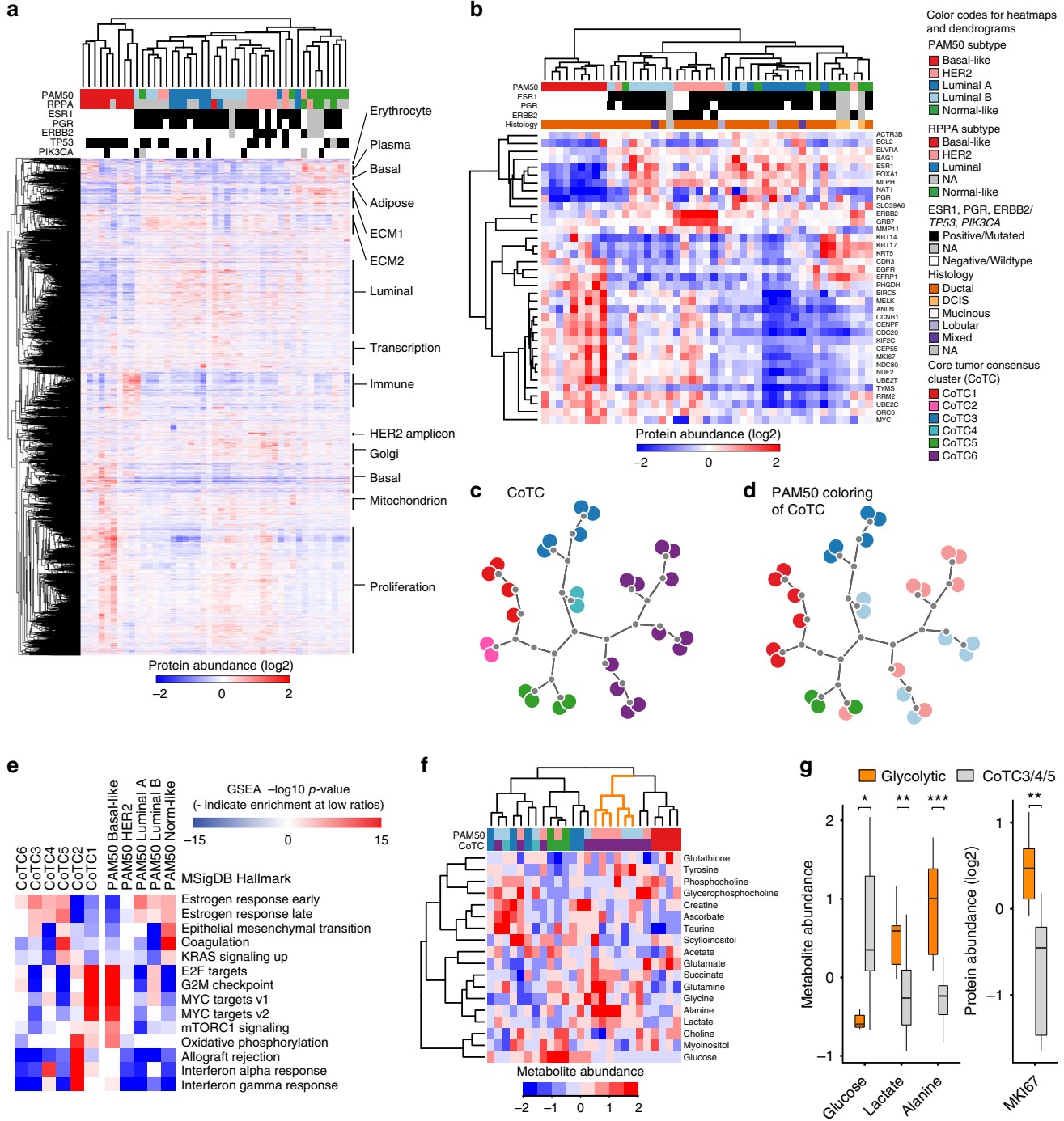

**Fig. 2** Proteome clustering, relation to PAM50 subtypes and metabolites. **a** Proteome-driven clustering of proteins mapping to 9995 gene symbols with overlapping quantification in all 45 tumors. Protein cluster characteristics, by GO enrichment analysis, are highlighted to the right (see Supplementary Fig. 2 for details). **b** Clustering of identified and quantified proteins from the PAM50 panel ($n = 37$). **c** Dendrogram visualization of core tumor consensus clusters (CoTC) into six clusters. For details, see Supplementary Methods and Supplementary Fig. 4. **d** PAM50 subtype assignments for the CoTCs in c. **e** Ranked gene set enrichment analysis (GSEA) of CoTC and PAM50 subtypes. **f** Clustering of HR-MAS measured metabolite levels and relation to CoTCs and PAM50 subtypes. Tumors with glycolytic characteristics are indicated in orange. HR-MAS data are not available for CoTC2 tumors. **g** Levels of glucose and its conversion product lactate and alanine, as well as MKI67 protein abundance in glycolytic tumors compared to other luminal tumors. T-test, *$p < 0.05$, **$p < 0.01$, ***$p < 0.001$. In box plots, center line represents median and the boxed region represents the first to third quartile, whiskers according to Tukey

Correlation also appears to be indicative of co-function, as mapping associations (defined by Pearson correlation > 0.5) of proteins marked by high variance across the Oslo2 Landscape cohort in a manner that minimizes edge length (protein nodes are in proximity to groups of nodes with which they share multiple edges, Supplementary Methods) illustrates that proteins functioning as components of similar biological processes are highly connected (Fig. 3c, Supplementary Fig. 7D, E); a feature also present in the CPTAC dataset (Supplementary Fig. 7F). Considering each CoTC and PAM50 group individually and

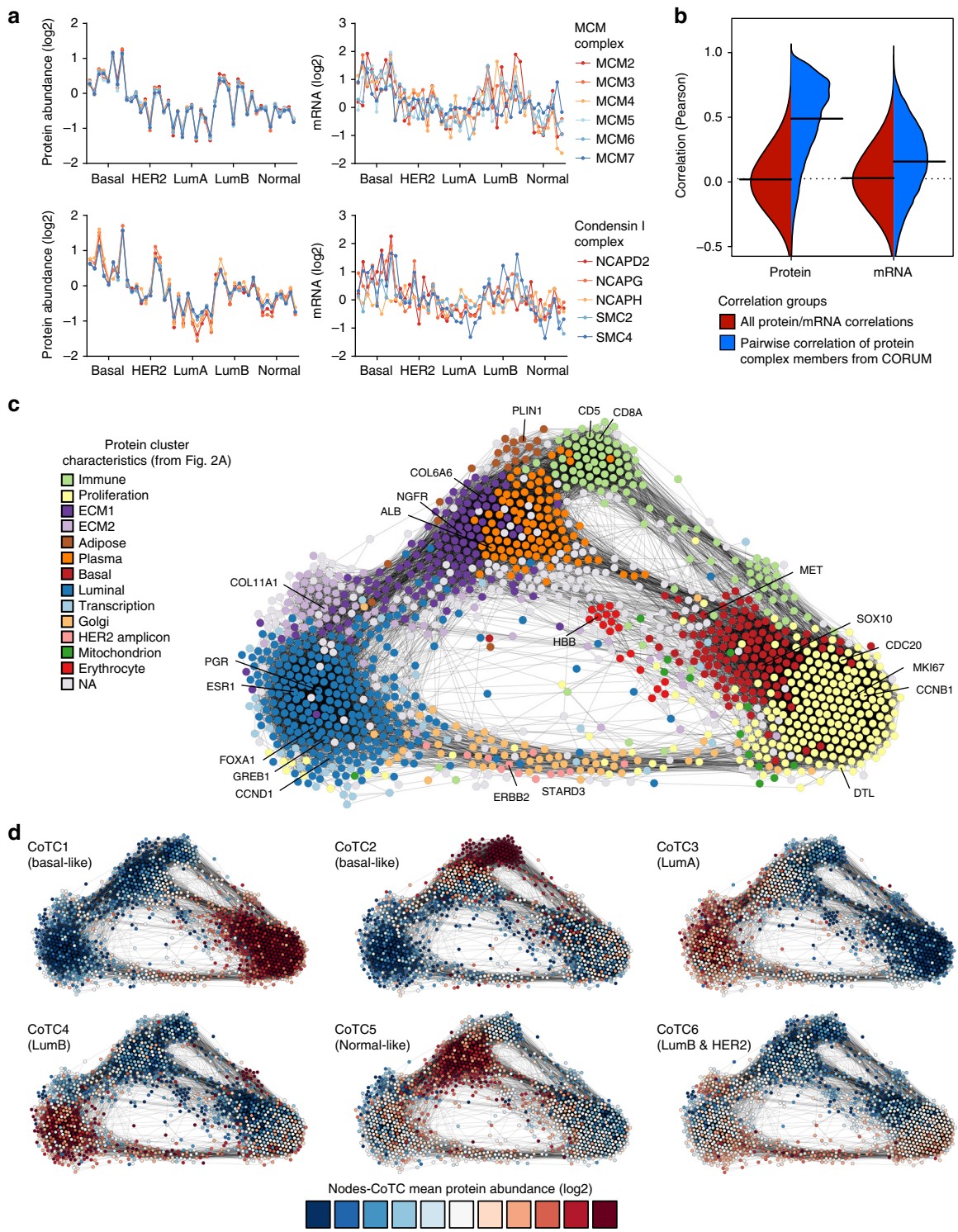

**Fig. 3** Proteome characteristics associate with tumor grouping. **a** Protein and RNA levels across tumors for known protein complexes (Supplementary Fig. 7A, B examples of more complexes). Basal indicates basal-like and normal indicates normal-like PAM50 subtype. **b** Comparison of all pairwise correlations to correlations from known interaction pairs from CORUM database, using quantitative protein and RNA levels across the 45 tumors (see Supplementary Fig. 7C for same analysis using Biogrid interactions). **c** Breast cancer protein correlation network based on 1447 high-variance proteins using > 0.5 Pearson correlation and KCore > 1 cutoff. Protein groups are defined and color coded based on GO enrichments in Fig. 2a, Supplementary Fig. 2, and 7D, E. **d** Visualization of average quantification of core tumor proteome consensus clusters (CoTC) in the correlation network. CoTCs are defined in Fig. 2c and Supplementary Fig. 4. Main PAM50 subtype(s) for each CoTC is indicated in parentheses

overlaying protein abundances onto this network provides snapshots of the defining characteristics of each group (Fig. 3d, Supplementary Fig. 8A–E), which are consistent with and expand upon the previously discussed enrichment analyses (Fig. 2e).

Abundances of proteins related to the immune response, including the MHC class (Supplementary Figs. 5D, 8F), are starkly elevated in CoTC2 as compared with CoTC1 tumors (all PAM50 basal-like). This may influence their depletion in proliferation-related proteins and suggests PAM50 basal-like may be an incomplete classification (consistent with Lehmann et al.[22]). CoTC3 (all luminal A) and CoTC4 (a subset of luminal B) tumors highly express luminal proteins, yet are distinguished by a small immune response network outpost enriched for function in the interferon alpha response (Figs. 2e, 3d) in CoTC4 members. Finally, CoTC5 tumors (primarily normal-like) are distinguished by elevated extracellular matrix cluster 1 (ECM1) and plasma protein abundances (Fig. 3d).

**MET and EGFR are coregulated in normal-like tumors**. Development of drug resistance is a nearly universal response to targeted cancer monotherapies and simultaneously inhibiting proteins in series or of parallel pathways is a promising treatment approach. Thus, we examined known drug targets for correlating expression (Fig. 4a, Supplementary Data 2), which would implicate them as operating in series/parallel and suggest that they are promising co-targets.

Protein abundances of ESR1, PGR, AR, and BCL2 are highly correlated (as measured by MS, Oslo2 Landscape cohort, $n = 45$, and RPPA, Oslo2 cohort, $n = 329$, and consistent with TCGA RPPA measurements, $n = 892$) (Fig. 4b), suggesting they may operate in concert and render tumors sensitive to simultaneous targeting by existing therapeutic estrogen, progesterone, and androgen hormone receptor inhibitors along with an apoptosis-inducing BCL2 antagonist (one of which recently received FDA approval)[23].

Similarly, protein abundances of MET and EGFR are highly correlated (Fig. 4a, c) and their co-expression may be a marker for basal-like (consistent with Kim et al.[24]) and normal-like tumors (Fig. 4d). Upon histopathological inspection in two independent cohorts (Oslo2 Landscape, $n = 40$, and Oslo1, $n = 530$), co-elevation of EGFR and MET appears to be confined to ductal carcinoma in situ (DCIS) regions for a subset of normal-like tumors (Fig. 4e–g, Supplementary Fig. 9, Supplementary Data 3), and high-resolution images of these regions in two tumors suggest EGFR-MET co-localization may confer an advantage for their in situ survival (Fig. 4h).

The normal-like subtype is often overlooked as a BC classification because of the tumors' close semblance to normal tissue and less aggressive luminal tumors. Identifying histopathological markers is an important step to ensuring patients are properly stratified into treatment regimens while revealing the inverse coregulation of EGFR and MET in invasive and DCIS regions provides fodder for therapeutic development within this understudied disease class.

**RNA–protein correlation analysis**. Transcriptomics has remained the standard-bearer in the molecular profiling of breast tumors since Perou et al.[1], first described the current PAM50 subtypes, and transcript quantity is widely used as a surrogate for protein abundance. Thus, we characterize the relationships between mRNA transcript and protein abundances to provide an understanding of when mRNA is a reliable surrogate for the protein product.

Positive and significant correlations exist across tumors between 70% of the proteins quantified in the Oslo2 Landscape cohort and their mRNA transcripts (Fig. 5a, Supplementary Data 4) (consistent with previous reports[5,7,25]) and do not appear dependent on protein/mRNA half-life (as measured in mice by Schwanhäusser et al.[26]), average protein precursor area, or number of peptide spectral matches (PSMs) (Supplementary Fig. 10A–D). However, proteins known to rapidly accumulate ubiquitin groups upon inhibition of the proteasome[27] have quantities significantly less correlated with their transcript abundances (Supplementary Fig. 10E), suggesting the influence of transcript regulation is buffered for those whose abundances are controlled at the protein level.

Additionally, structural ribosomal proteins and those of the inner-mitochondrial membrane embedded electron transport chain (oxidative phosphorylation) are not as highly correlated with transcript quantity as are groups of soluble metabolic (amino acid metabolism, fatty acid metabolism, and steroid hormone synthesis) and signaling/proliferation-related (estrogen and interferon responses, MTORC1 signaling, E2F targets, and G2M checkpoint) proteins (Supplementary Fig. 10F–G, Supplementary Data 4).

Strikingly, protein products of transcripts profiled as part of BC prognostic panels are significantly enriched for high mRNA to protein correlations (Fig. 5b). This suggest that the robustness of clinical mRNA markers is due, at least in part, to being reliable protein surrogates and demonstrates a clear link to protein phenotype. However, overall mRNA–protein correlation appears to have a non-linear dependence on variability (Supplementary Fig. 11A, C). Modeling this dependence (Supplementary Fig. 11B, D, Supplementary Data 4, Supplementary Equation 1–4) still reveals prognostic mRNA signatures to have higher correlation with their protein products than expected (Supplementary Fig. 11E–R).

Genes causally associated with cancer (COSMIC)[28] and breast cancer[29] display varied mRNA–protein correlations, indicating that some proteins should not be studied by mRNA expression serving as a surrogate measurement (Fig. 5c).

Highly proliferative tumors (basal-like, HER2, luminal B, and/or high MKI67) have a tendency to have more correlated proteomes and transcriptomes than lowly proliferative tumors (luminal A, normal-like) (Supplementary Fig. 12). This is further supported by high abundances of proteins involved in transcription, splicing, translation, and cell cycle being associated with high-tumor mRNA–protein correlation (Fig. 5d). On the other hand, high abundances of the extracellular matrix and plasma proteins are linked to a poor tumor mRNA–protein correlation (Fig. 5d).

**Impact of CNAs is dampened at the protein level**. Much of contemporary thinking labels cancer as a disease of the genome, and gene copy-number alterations (CNAs) are known to be associated with expression of their corresponding transcripts (i.e., associated in *cis*) in breast tumors[30]. Thus, we explore whether variance across proteomes in the Oslo2 Landscape cohort can be accounted for by CNAs.

Consistent with Curtis et al.[30], mRNA expression in *cis* is associated with CNAs, and the same effect, though dampened, appears to be present at the protein level (Supplementary Fig. 13A). Imposing fold change and Wilcoxon test statistic cutoffs (Supplementary Fig. 7B–E) allows for determination of genes with significant CNA (gain or loss) to mRNA or protein associations (Supplementary Data 5). Considering gain and loss effects, a gene selected whose protein abundance is associated with a *cis* CNA is twice as likely to have both mRNA expression and protein abundance associated with that CNA than a gene selected based on an mRNA–CNA association (Fig. 6a, b), further

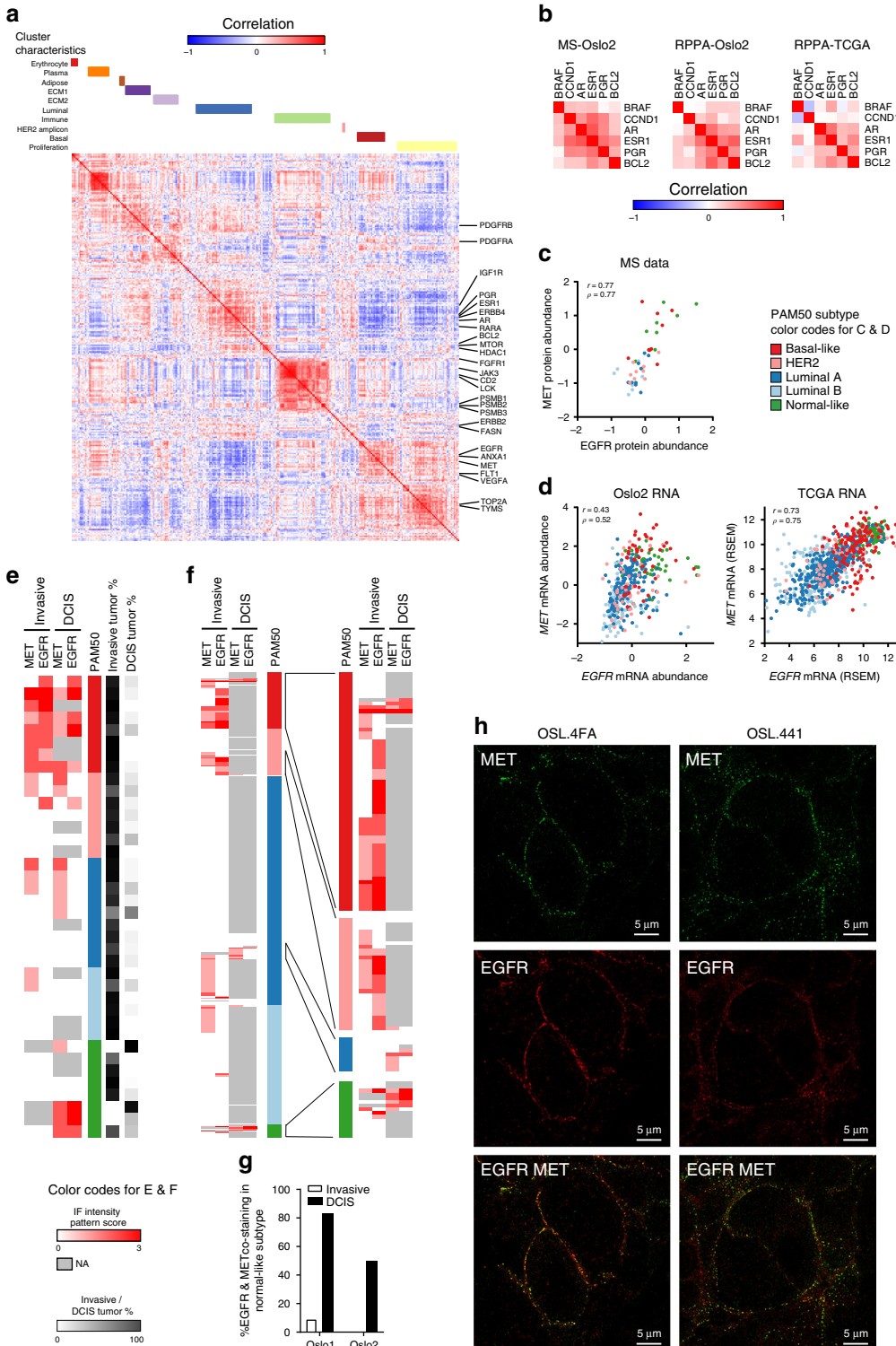

**Fig. 4** Druggable proteome analysis. **a** Correlation matrix of all 290 FDA approved drug targets detected and quantified across all 45 tumors. Top panel shows the connection to annotated protein clusters defined in Fig. 2a, Supplementary Fig. 2. Selected BC targets and potential targets, highlighted on the right side. **b** Correlation matrices, comparing MS data, and antibody-based quantification (RPPA), from Oslo2 ($n = 329$) and TCGA ($n = 892$), for correlating luminal drug targets from **a** identified in all three datasets. **c** Scatter plot of EGFR and MET protein levels in Oslo2 MS data. **d** Scatter plot of *EGFR* and *MET* mRNA levels in the whole Oslo2 cohort ($n = 378$) and TCGA ($n = 950$). Correlation coefficients are indicated as Pearson's $r$ and Spearman's $\rho$ in **c** and **d**. **e** Scoring of EGFR and MET IF staining pattern from whole sections of 40 of the Oslo2 tumors analyzed by MS proteomics. Tumors are arranged according to PAM50 subtype and separated by invasive and ductal carcinoma in situ (DCIS) tumor regions. See Supplementary Fig. 9 for staining examples. **f** Scoring of IF staining pattern from Oslo1 cohort ($n = 530$) in the same way as in **e**. **g** Co-staining of EGFR and MET in the normal-like subtype. Evaluable DCIS and invasive components from **e** and **f** are shown. **h** Super-resolution STED microscopy of EGFR and MET staining in in situ regions of two normal-like tumor

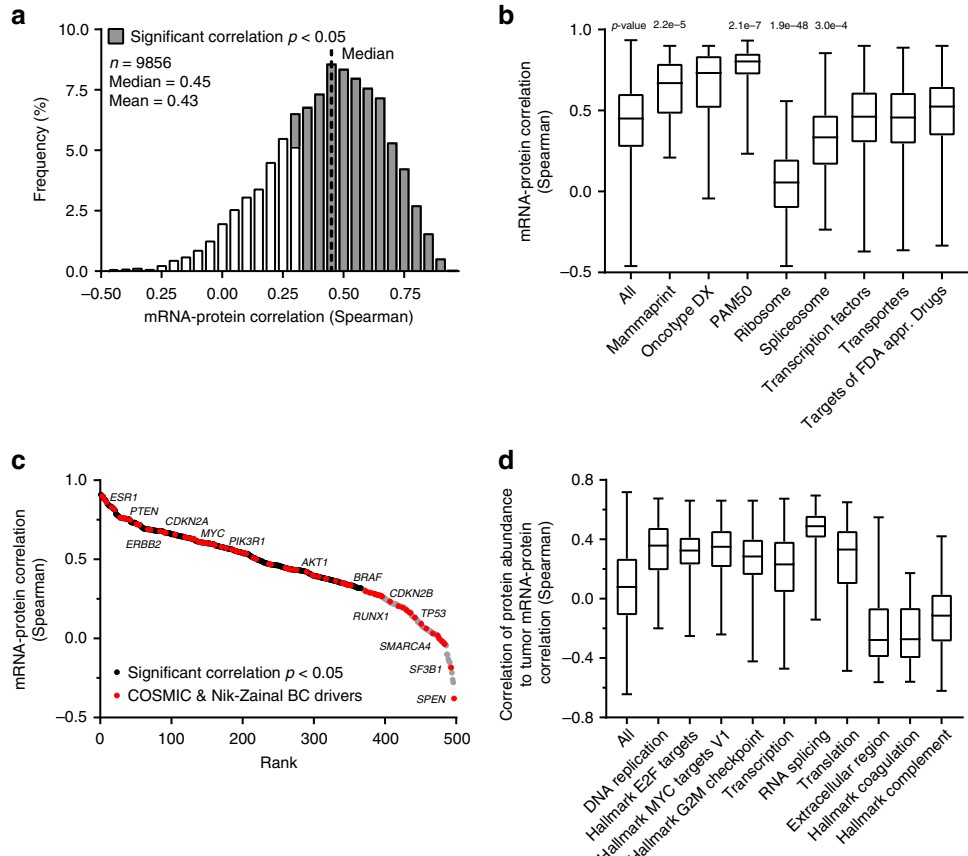

**Fig. 5** RNA–protein correlation analysis. **a** Correlation between protein and mRNA quantitative values (Spearman) of individual genes. **b** Distribution of mRNA–protein correlations for selected groups of genes. Gene groups were compared with all correlations using Mann–Whitney U test. For additional gene groups and mRNA–protein correlation analysis considering data distribution, see Supplementary Fig. 10, 11, and Supplementary Data 4. **c** Ranked mRNA–protein correlations for genes causally associated with cancer (COSMIC)[28] and breast cancer (Nik-Zainal)[29]. **d** Gene ontology and hallmarks enriched at the top or bottom of proteins associated with tumor mRNA–protein correlation. All visualized protein groups have a p-value enrichment below 1E-17 using Mann–Whitney U test. In box plots, center line represents median and the boxed region represents the first to third quartile, whiskers indicate the maximum and minimum values

supporting the hypothesis that CNA effects are attenuated at the protein level.

The attenuation of CNA effects may be explained by regulatory mechanisms at the protein level. To investigate, genes are split into lowly and highly attenuated groups based on a gaussian mixture model of protein abundance and mRNA expression correlations to CNAs. The gene subset with high mRNA–CNA and low protein–CNA correlations are defined as highly attenuated (Fig. 6c), and accumulate ubiquitin (according to Kim et al.[27]) at an elevated rate (Fig. 6d); suggesting the abundances of these highly attenuated proteins are more regulated at the protein level (consistent with Gonçalves et al.[31]).

CNAs appear to be distributed over the genome (Fig. 6e, Supplementary Fig. 13F), though many gain effects (mRNA and protein) are localized to chromosome 17, and cover genes of the HER2 amplicon, while loss effects influence estrogen signaling (Fig. 6f) and are concentrated in basal tumors (Supplementary Fig. 13G). Interestingly, six genomic losses (including that of CCNB1) are associated with increased abundances at both the mRNA and protein level (Supplementary Fig. 13F, Supplementary Data 5), suggesting possible implementation of a compensatory mechanism. Of note, Myhre et al. observed similar effects of CCNB1 gene copy-number loss[32].

Curtis et al.[30] classifies breast tumors (n = 2000) based on cis associations of CNAs and mRNA expression of 619 genes. Of those that we quantified at the protein level in the Oslo2 Landscape cohort, 83% overlap with the set of genes having significant copy number to transcript or to protein abundance associations as determined by the linear regression method implemented by Curtis et al.[30] (Fig. 6g). This consistency demonstrates that a cohort sized for a tractable high-quality proteome quantification study contains sufficient statistical power to reproduce findings from much larger cohorts.

**Proteogenomics identifies candidate immunotherapeutic targets.** Translation of tumorigenic genomic aberrations produces tumor-specific proteins, whose immunoreactivity renders them ideal candidate antigens for targeted immunotherapies. Thus, we apply our recently developed integrated proteogenomics analysis workflow[11,12] to the in-depth proteome characterization of the Oslo2 Landscape cohort.

In brief, MS spectra are searched against databases of known peptides, SNPs, mutations, and theoretical peptides from genomic regions believed to be noncoding derived from the six reading frame translation of the entire genome (restricted based on peptide isoelectric point). Spectra matching known peptides and multiple genomic regions are filtered out along with single amino acid variants (SAAVs) not meeting stringent verification criteria imposed by SpectrumAI[12] (Fig. 7a).

Among the Oslo2 Landscape cohort, hundreds of peptides are identified mapping to genomic regions thought to be noncoding or intronic (pseudogenes, noncoding RNA), or corresponding to

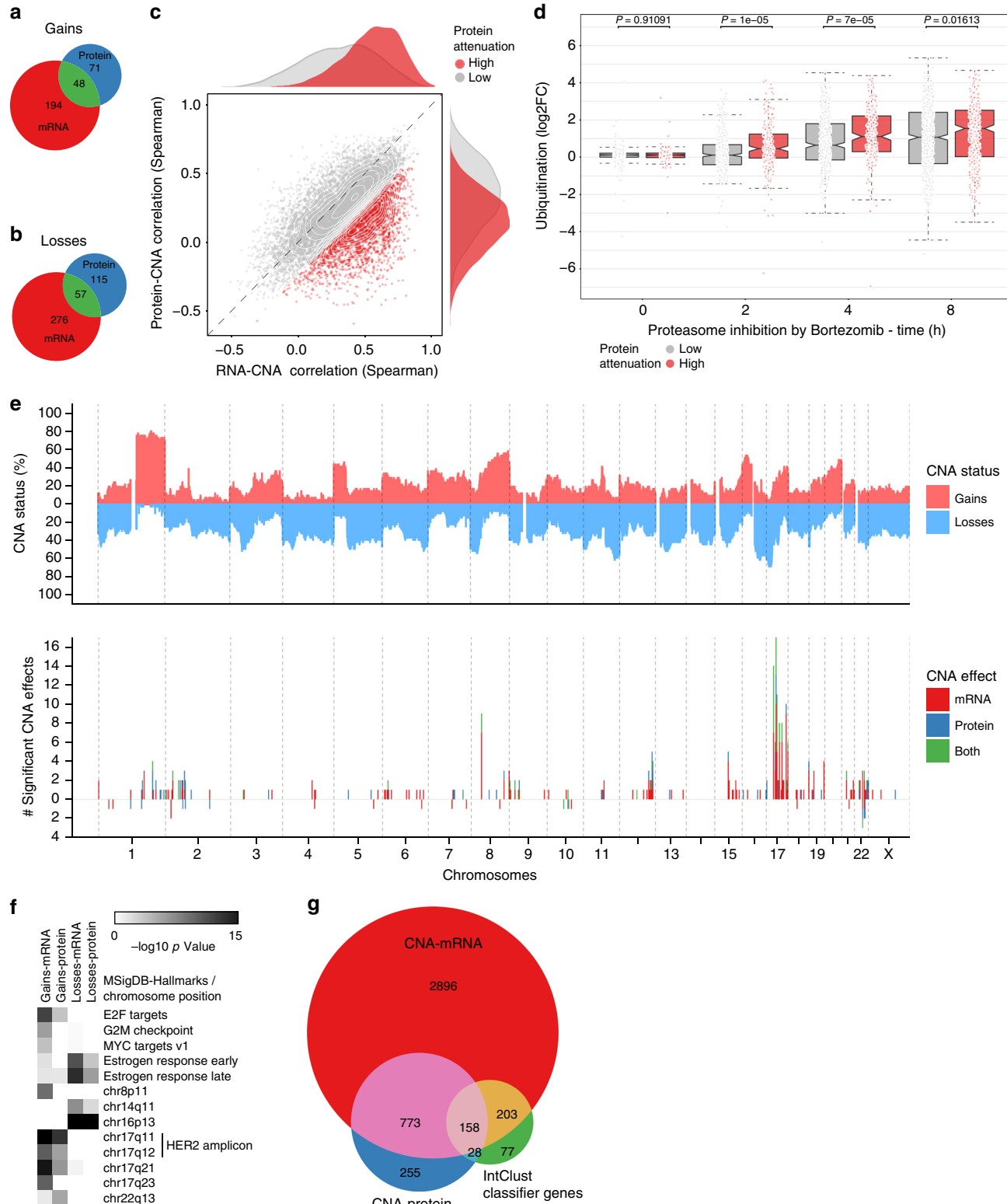

**Fig. 6** Gene copy number effects on mRNA and protein levels. **a**, **b** Venn diagrams displaying CNAs associated with mRNA and/or protein levels for **a** gains and **b** losses in *cis*. See Supplementary Fig. 13 and Supplementary Methods for defining the CNA–mRNA/protein associations. **c** Scatter plot of CNA correlation to RNA and protein to identify CNA effects attenuated at the protein level. Attenuated proteins (in red) were identified using a Gaussian mixture model. **d** Boxplot of ubiquitinylation site fold change following bortezomib proteasome inhibition for proteins defined as attenuated in panel **c** (red) compared with non-attenuated (gray). Wilcoxon test was used on the ubiquitinylation data from Kim et al.[27]. **e** Genomic distribution of CNAs and CNA effects of gains from **a**. **f** MsigDB and chromosome position enrichment analysis of CNA effects on mRNA and protein from **a** and **b**. Hypergeometric test. **g** Overlap of CNA effects to IntClust classifier genes (466 of 619 genes overlapped all three datasets)[30]. CNA effects associated with ANOVA as in Curtis et al.[30]

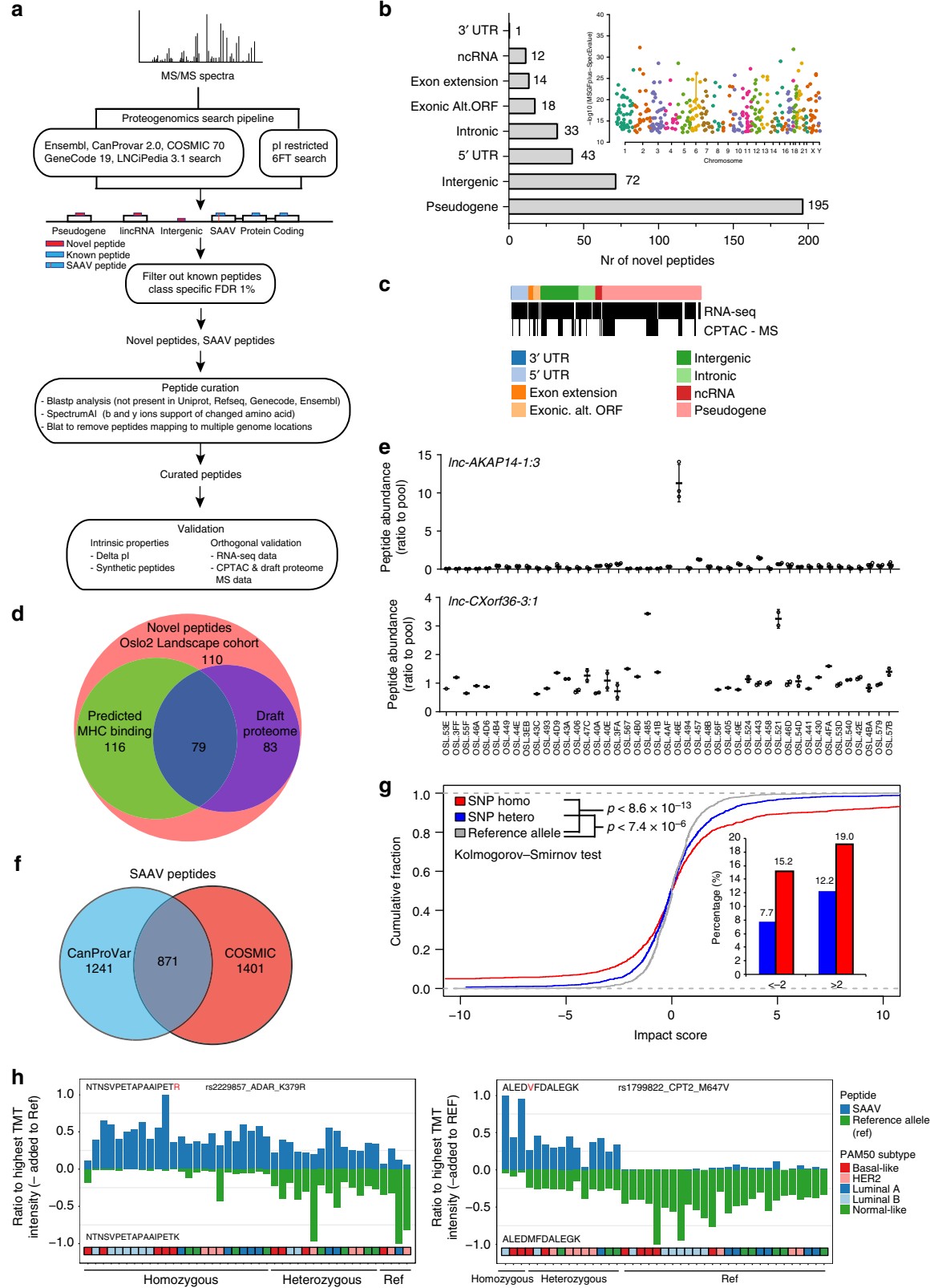

un-annotated alternative gene translations (exon extensions, 3′ and 5′ UTRs, exonic alternate open-reading frames) (Fig. 7b, Supplementary Data 6); two-thirds of which are identified by at least two PSMs (Supplementary Fig. 14A). Furthermore, these peptides have a similar pI distribution as known peptides (Supplementary Fig. 14B), and 10% of the corresponding coding

genomic loci are supported by mappings of at least two peptides (Supplementary Fig. 14C). Additionally, RNAseq measurements and proteomic MS-spectra[5,33] verify that many of these peptides are transcribed and translated in independent breast tumor cohorts (Fig. 7c, Supplementary Fig. 14D), while CAGE[34] and ribosomal profiling[35] reveal they are transcribed and translated in

**Fig. 7** Proteogenomics analysis. **a** Overview of the proteogenomics workflow and additional data levels used for validation. **b** Curated peptides from novel coding regions. Categories according to genome annotation in the respective loci. Inset shows Manhattan plot of novel peptide distribution across the human genome. **c** Orthogonal evidence of novel peptides by public domain data, indicated by the presence of black bars in corresponding rows for RNA-seq[33], and re-analysis of proteomics data on breast tumors[5]. See Supplementary Fig. 14 for details. **d** Prediction of MHC class I binding[36] and identification in normal tissues from draft proteome data[37] among novel peptides. **e** High levels of novel peptides from lncRNA *lnc-AKAP14–1:3* in one Luminal A (top) tumor and in two tumors (Luminal A and B) for *lnc-CXorf36–3:1* (bottom). **f** Unique and overlapping identifications of curated SAAV peptides from CanProVar and COSMIC databases. **g** Impact of SNPs (from iCOG array), with corresponding SAAV peptide identification, on protein levels. Impact score is plotted cumulatively for reference allele, hetero and homozygous SNPs. Percentage of impact scores below −2 and above 2 are shown in the inset. See Supplementary Fig. 15B for examples. **h** Allele-specific protein levels displaying SAAV peptide and matched reference allele peptide quantification cross the 45 tumors. Peptide quantification is categorized into reference allele (Ref), hetero- and homozygous SNPs, based on iCogs data. See Supplementary Fig. 15C for more examples

other systems (normal and cancer) (Supplementary Fig. 14E, F). Moreover, genetic loci corresponding to peptides previously annotated as pseudogenes and noncoding RNA are more highly conserved than random loci with the same annotation (Supplementary Fig. 14G). Finally, 61 peptide identifications are verified by MS-spectra of synthetic peptides (out of 67 attempted) (Supplementary Data 7).

Of the identified peptides, 30% (116) were predicted to bind MHC class I[36] and not identified in MS data from normal tissue[37] (Fig. 7d, Supplementary Data 6). Patient-specific candidate immunotargets are exemplified by *lnc-AKAP14–1:3* and *lnc-CXorf36–3:1*; each corresponding to noncoding regions whose protein products are respectively elevated in one and two tumors, and whose presence is supported by multiple mapped peptides (Fig. 7e). Tumor specificity in the breast of proposed immunotargets is evidenced by their absence in normal surrounding tissue as detected in a separate MS proteomics assay (Supplementary Fig. 14H, Supplementary Data 8). These profiles are contrasted to those which suggest patient (rather than tumor) specificity (Supplementary Fig. 14I, Supplementary Data 8).

Identifying antigen targets in the proteome, as opposed to the transcriptome, may streamline development of immunotherapies, because the immune system is activated by protein fragments displayed on the MHC. These high-confidence identifications demonstrate the prominent role proteogenomic analyses of high-quality proteome spectra will have in the push towards individualized medicine.

**Single amino acids variants impact protein abundances**. Certain SAAVs of proteins, naturally occurring or acquired via somatic mutations, are known risk factors for the development or prognosis/therapy response predictors of cancer. Cancer-related SAAVs are cataloged in the CanProvar and COSMIC databases. Thousands of these SAAVs match Oslo2 Landscape proteome spectra (after applying the stringent SpectrumAI filter) (Fig. 7f, Supplementary Data 6), have a similar ΔpI distribution as known peptides (Supplementary Fig. 15A), and 28 (out of 31 randomly selected) spectral identifications are confirmed with synthetic peptides (Supplementary Data 7). Furthermore, breast cancer driver genes[29], including *MAP3K1*, *AKT2*, *FOXA1*, *ERBB2*, and *CDKN1B*, are amongst proteins identified with SAAVs (Supplementary Data S6).

## Discussion
Molecular characterization has progressively stratified breast cancer patients into more disease-type specific cohorts; with the first clinical manifestation being the adoption of immunohistochemical evaluation of ER, PR, KI67, and later HER2 expression as primary determinants of treatment regimens. Though interpretation of marker expression and administration of targeted therapies denotes a vast improvement over sole reliance on

staging and grade, variances in specimen preparation and a heavy reliance on human judgment along with technological advances in measuring gene expression fostered development of unbiased whole-transcriptome profiling as an accompaniment.

Importantly, unbiased analyses of thousands of gene transcripts largely recapitulates pathological marker classifications, inspiring confidence that they could further delineate patients responsive to targeted therapies. Indeed, transcript profiling of gene panels selected to represent the variance across tumor subtypes is recommended for assigning treatment courses for patients with early stage ER+ /HER2− tumors[2,3] and may prevent patients from receiving chemotherapy who do not stand to benefit[38]. Furthermore, mRNA profile-based stratification has defined a normal-like subtype that clinical markers alone do not recognize.

Proteome characterization has provided valuable insight on CNA effects and their attenuation at the protein level[4,5]. However, to date, unbiased proteome profiling has not resolved a consistent variance with mRNA profiling across known breast tumor subtypes. This has largely been accredited to the unreliability of mRNA as a surrogate for protein abundance. However, given unbiased mRNA profiling distinguishes function and differential responses to therapy; adherence to the tenets of the central dogma dictates proteome profiles would distinguish the same differences while layering additional insights. In fact, our findings suggest expression profiles of prognostic mRNA panels stratify breast tumors based on known biological variances partly because the selected transcripts are reliable protein surrogates.

Herein, we present a landmark study, which is the first to recapitulate known function-enriched stratifications of breast tumors based on unbiased analyses of proteome profiles. We attribute this to the proteome coverage and high-quality quantification resulting from the reduction in peptide fraction complexity accomplished by HiRIEF separation[11]. The reliability of protein quantification is attested to by the remarkable correlation of protein complex members, and suggests protein co-function may be inferred by high cross-tumor correlation. From here, we layer additional insight over the breast cancer landscape by further analyses of proteomes and parallel systems measurements. Such insights include identifying a definitive separation of basal-like tumors based on immune components, postulating regulatory control of characteristic cancer hallmark genes by phosphorylation of specific groups of regulatory proteins, revealing the glycolytic preference of proliferative tumor subtypes, and determining copy-number alterations may be attenuated by protein-level regulatory mechanisms, such as ubiquitinylation and subsequent degradation in the proteasome. Furthermore, we validate the often ambiguous mRNA-based normal-like subtype, both with an unbiased approach and the identification of MET-EGFR coexpression as a biomarker; potentially for the DCIS component.

With an eye toward advancing to individually tailored therapies, we apply our recently developed proteogenomics pipeline[12]

and identify proteins in individual tumors, with high confidence, corresponding to un-described gene variants, noncoding regions, or regions thought to be noncoding due to poor genome annotation. We postulate protein products of undescribed gene variants and noncoding regions are the consequences of cancer genome instability, and that they are strong candidates of tumor-specific targets for immunotherapies.

As breast tumors are continuously revealed to be individually unique diseases, considerations of molecular profiles will become paramount in selecting from available treatment options and developing new ones. Though mRNA profiling has been initially dominant in this role, the "landscape" study presented herein demonstrates the instrumental contribution analyses of the quantitative proteome will have moving forward. Patient stratification based on high-quality proteome MS data is marked by consistency with multiple systems level and immunohistochemical readouts, underscoring the utility of a multi-faceted approach to translate systems level findings into effective therapeutic strategies.

We have created a user friendly and easily accessible data portal with analysis tools to ensure that this rich dataset can be explored by the research community, available at: www.breastcancerlandscape.org.

## Methods

**HiRIEF-nanoLC-MS/MS-based proteomics and proteogenomics.** Tumor samples from Oslo2 cohort were prepared for MS analysis using a modified version of the spin filter-aided sample preparation protocol[11,39], and peptides were separated using immobilized pH gradient-isoelectric focusing (IPG-IEF) on narrow range pH 3.7–4.9 and 3–10 strips[11]. Peptide fractions were separated using a 3000 RSLCnano system and analyzed using a Thermo Scientific Q Exactive. MSGF + Percolator in the Galaxy platform was used to match MS spectra to the Ensembl 75 human protein database[15,16]. Protein identifications were limited to 1% protein FDR[40]. Proteogenomics was performed as described by Branca et al. and Zhu et al.[11,12]. See Supplementary information for additional methods and details.

**Reporting summary.** Further information on experimental design is available in the Nature Research Reporting Summary linked to this article.

## Data availability

The MS data have been deposited in the ProteomeXchange database under the accession code PXD008841 and PXD011385. The additional datasets referenced during the study are available in public repositories and can be found in the Data availability table in Supplementary information. All the other data supporting the findings of this study are available within the article and its Supplementary information files and from the corresponding author upon reasonable request.

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

## Acknowledgements

We acknowledge support from NBIS, BioMS, and SciLifeLab proteogenomics core facilities. J.L. is supported by the Swedish Research Council, the Swedish Cancer Society, the Swedish Foundation for Strategic Research, the Stockholm County Council (ALF),

the Cancer Society in Stockholm, and AstraZeneca Smart Trial grant. G.B.M. is supported by NCI grants CA016672, U24CA209851, U24CA210949, U24CA210950 and Komen grant SAC110052, and BCRF-16–109 from the Breast Cancer Research Foundation. N.M.V. is supported by the Wenner-Gren Foundation and the Swedish Cancer Society. B.S. is supported by the Knut and Alice Wallenberg Foundation. S.N. & E.K.H. are financed by the Norwegian Regional Health Authorities (2014061). We thank Eldri U Due for sample preparation. D.C.J. and H.B. are supported by National Microscopy Infrastructure, NMI (VR-RFI 2016–00968).

## Author contributions

Conceptualization: H.J.J., J.L., A.L.B.D., K.K.S., and G.B.M.; tumor resource, OSBREAC: A.L.B.D., K.K.S., E.S., Ø.G., T.S., T.L., and V.N.K.; MS tumor investigation: H.J.J., M.W., R.M.B., and J.B.; Software, visualization, formal analysis of MS data for correlation analysis, correlation network and subtype: A.F., H.J.J., E.F., L.M.O., and M.R.A.; RPPA tumor resource, investigation, and formal analysis: M.H.H., W.Z., G.B.M., H.J.J., and M.R.A.; Metabolomics investigation, resource and formal analysis: N.M.V., T.H.H., T.F.B., and H.J.J.; pathology investigation, formal analysis and resource: F.S., E.B., Ø.G., T.S., E.B., and H.G.R.; super-resolution microscopy investigation and formal analysis: F.S., D.C.J., and H.B.; software and formal analysis of mRNA–protein correlation: B.S. and H.J.J.; resource, software and formal analysis of copy-number association to mRNA and protein: I.S., O.L., E.K.H., S.N., and H.J.J.; Proteogenomics software, investigation, and formal analysis: Y.Z., H.J.J., R.M.B., and M.H.; database investigation: A.F; writing—original draft: H.J.J., N.M.V., and J.L.; Writing—review and editing: all authors. Project administration: H.J.J., J.L., M.H.H., A.L.B.D., and K.K.S., and project Supervision: H.J.J., J.L., A.L.B.D., and K.K.S.

## Additional information

**Competing interests:** The authors declare no competing interests.

Henrik J. Johansson [1], Fabio Socciarelli[1], Nathaniel M. Vacanti[1,29], Mads H. Haugen[2], Yafeng Zhu[1], Ioannis Siavelis[1], Alejandro Fernandez-Woodbridge[1], Miriam R. Aure[3], Bengt Sennblad[4], Mattias Vesterlund [1], Rui M. Branca [1], Lukas M. Orre[1], Mikael Huss[5], Erik Fredlund[1], Elsa Beraki[6], Øystein Garred[6], Jorrit Boekel[1], Torill Sauer[7,8], Wei Zhao[9], Silje Nord[3], Elen K. Höglander[3], Daniel C. Jans[10], Hjalmar Brismar[10,11], Tonje H. Haukaas[12], Tone F. Bathen[12], Ellen Schlichting[13], Bjørn Naume[8,14], Consortia Oslo Breast Cancer Research Consortium (OSBREAC), Torben Luders[8,15], Elin Borgen[6], Vessela N. Kristensen[3,8,15], Hege G. Russnes[3], Ole Christian Lingjærde [3,16], Gordon B. Mills[9], Kristine K. Sahlberg[3,17], Anne-Lise Børresen-Dale[3,8] & Janne Lehtiö [1]

[1]Science for Life Laboratory, Department of Oncology-Pathology, Karolinska Institutet, 171 21 Solna, Sweden. [2]Department of Tumor Biology and Department of Cancer Genetics, Institute for Cancer Research, Oslo University Hospital, 0424 Oslo, Norway. [3]Department of Cancer Genetics, Institute for Cancer Research, Oslo University Hospital, 0424 Oslo, Norway. [4]Department of Cell and Molecular Biology, National Bioinformatics Infrastructure Sweden, Science for Life Laboratory, Uppsala University, 752 37 Uppsala, Sweden. [5]Department of Biochemistry and Biophysics, National Bioinformatics Infrastructure Sweden, Science for Life Laboratory, Stockholm University, 171 21 Solna, Sweden. [6]Department of Pathology, Oslo University Hospital, 0424 Oslo, Norway. [7]Department of Pathology, Akershus University Hospital, 1478 Lørenskog, Norway. [8]Institute for Clinical Medicine, University of Oslo, 0318 Oslo, Norway. [9]Department of Systems Biology, The University of Texas MD Anderson Cancer Center, Houston, TX 77230-1429, USA. [10]Department of Applied Physics, KTH Royal Institute of Technology, 171 21 Solna, Sweden. [11]Department of Womens's and Children's Health, Karolinska Institutet, 171 21 Solna, Sweden. [12]Department of Circulation and Medical Imaging, The Norwegian University of Science and Technology – NTNU, 7491 Trondheim, Norway. [13]Section for Breast- and Endocrine Surgery, Department of Cancer, Division of Surgery, Cancer and Transplantation Medicine, Oslo University Hospital, 0424 Oslo, Norway. [14]Department of Oncology, Division of Surgery and Cancer and Transplantation Medicine, Oslo University Hospital, 0424 Oslo, Norway. [15]Department of Clinical Molecular Biology and Laboratory Science (EpiGen), Division of Medicine, Akershus University Hospital, 1478 Lørenskog, Norway. [16]Centre for Cancer Biomedicine, University of Oslo, 0424 Oslo, Norway. [17]Department of Research, Vestre Viken Hospital Trust, 3004 Drammen, Norway. [29]Present address: Cornell University, Division of Nutritional Sciences, Ithaca, NY 14853, USA. A full list of consortium members appears at the end of the paper.

## Consortia Oslo Breast Cancer Research Consortium (OSBREAC)

Jürgen Geisler[8,18,19], Solveig Hofvind[20,21], Olav Engebråten[8,14,22], Gry Aarum Geitvik[23], Anita Langerød[23], Rolf Kåresen[8,24], Gunhild Mari Mælandsmo[22,25], Therese Sørlie[23], Helle Kristine Skjerven[26], Daehoon Park[27] & Olaf-Johan Hartman-Johnsen[28]

[18]Department of Oncology, Akershus University Hospital, Lørenskog 1478, Norway. [19]Division of Medicine, Akershus University Hospital, Lørenskog 1478, Norway. [20]Cancer Registry of Norway, Oslo 0379, Norway. [21]Oslo and Akershus University College of Applied Sciences, Faculty of Health Science, Oslo 0130, Norway. [22]Department of Tumor Biology, Institute for Cancer Research, Oslo University Hospital, Oslo 0379, Norway. [23]Department of Cancer Genetics, Institute for Cancer Research, Oslo University Hospital, The Norwegian Radium Hospital, Oslo 0379, Norway. [24]Department of Breast- and Endocrine Surgery, Division of Surgery, Cancer and Transplantation, Oslo University Hospital, Oslo 0379, Norway. [25]Department of Pharmacy, Faculty of Health Sciences, University of Tromsø, Tromsø 9010, Norway. [26]Breast and Endocrine Surgery, Department of Breast and Endocrine Surgery, Vestre Viken Hospital Trust, Drammen 3004, Norway. [27]Department of Pathology, Vestre Viken Hospital Trust, Drammen 3004, Norway. [28]Østfold Hospital, Østfold 1714, Norway

