## [Peer Review File · Nature Communications]

Reviewers' Comments:

Reviewer #1:

Remarks to the Author:

I really liked the analysis linking the peptides to the non-coding genome and would have like to see some expanded analyses and focus on that aspect of the paper. Were there specific peptides associated with ER/PR/HER2 status or the RNA/Protein classifications. I also didn't understand the bars in Figure 7C. The legend referred to Figure S8 for details but I'm not sure how that helps. Do the black lines represent peptides found in the RNAseq and the MS from CPTAC? Why are so few found in the MS data?

Why were the proteins restricted to the 9,995 genes that were present in all 45 tumors? There could be subtype-specific protein expression that might be incredibly useful for classification.

I looked at the centroid correlations for your 45 tumors and while the Basal-like and Luminal A are very strongly those subtypes, the selections of tumors for the other subtypes are not really that strong as shown in Table S1 and Figure S2A. Some of the correlations for the HER2 samples were really low and for the normal-like there were a few that were equally high for both normal-like and Luminal A. It seems a bit unfair to say that RNA classification was poorer when these were not really prototypical samples to start with. CoTC1 and CoTC2 wouldn't separate in the PAM50 analysis because the set of genes was enriched for tumor-intrinsic genes verses non-tumor genes such as the immune proteins that separate those groups. Based on the CoTC dendrogram visualization in Figure 2C, it seems like group 6 should have been separated into 2-3 groups. Is there concern with the limited sample numbers after reducing to the subset of samples, that the clustering is more driven by individual differences. The CPTAC data looks more like 3 groups than 6.

For figure 4E and F, would it be better to have the image show the correlation between MET and EGFR expression than the correlation for each with its respective RNA since you already showed they were similar? It might help with the interpretation of 4E and F.

In Figure 5B, is the lower correlation with ribosome and spliceosome more a function of difficulty in mapping RNA transcripts due to either highly similar genes (ribosomal) or smaller genes (spliceosome) and therefore the RNA quantification is not as accurate in those genes?

For figure 5C, is the poor tumor-protein correlation at all linked to tumor purity levels? The samples with the lower overall correlation also had higher non-tumor proteins expressed. If the RNA and Protein were from separate pieces of tumor, the cellular heterogeneity could also account for the poorer correlation. How were the proteins selected to show in 5C?

Minor comments:

- Figure 3A says correlation between protein and RNA, but you aren't showing correlation. Just the plots of the RNA and Protein separately.
- Figure S1, what is a TMT10 set?
- Figure S8, what is the ranking of the peptides? Also this figure has a legend for K, but only goes to J.
-

Reviewer #2:

Remarks to the Author:

In this manuscript, Johansson et al. performed a proteomic analysis of 45 breast tumors, including 9 from each of the five PAM50 subtypes. Because these tumors were previously analyzed by other omics platforms, they further performed integrative analyses by combining proteomics data with

other published data.

The main strength of the study includes the depth and quality of the proteomic profiling data and a large number of integrative analyses.

However, there are several major weaknesses. First, the key contribution of the study is not clear. Second, the sample size is very small, especially for subtype analysis. Third, some quantitative statements are not supported by statistical analysis.

1. This study is very similar to the CPTAC breast cancer study (Nature, 2016), and most of the findings have been previously reported in this and other proteogenomic studies, including: 1) proteomic subtypes overlap with transcriptomic subtypes but provide novel information; 2) impact of DNA copy number alterations is dampened at the protein level; 3) identification of novel and SAAV peptides. There are some potentially interesting findings, but they are mixed with other results, and thus the major contribution of the manuscript is not clear. It would be very useful to separate new findings from results that simply confirm previous findings. The manuscript spends too much space on the latter, and the potentially interesting stories are not fully developed. For example, it would be nice to have experimental data to evaluate the functional impact of co-targeting MET and EGFR.
2. Multiple methods were used to derive sample clusters based on proteomics data, including clustering based on all genes, clustering based on PAM50 genes, subtyping/consensus clustering based on selected genes and samples. This is very confusing. It is not clear which classification best represents proteomic heterogeneity.
3. Only 29 samples were used for subtyping/consensus clustering. The sample size is very small for subtype analysis. Moreover, these samples were divided into 6 classes (CoTCs), and most subtypes included only a few samples. The authors highlighted CoTC2 as a new finding, but it only included 2 samples. A subtype with only 2 samples is very questionable.
4. The authors tried to validate their proteomic subtypes in the CPTAC dataset and claimed that "Similar results were obtained when applying the unsupervised clustering on CPTAC MS-based quantified breast tumor proteomes". However, this was not supported by any quantitative evaluation. Actually, Figure S3I and S3J look very different to me. For example, Luminal A and Basal samples are perfectly clustered in S3I but not S3J.
5. In order to formally use CPTAC data as a validation set, one possibility is to come up with a classifier from the current dataset and then apply the classifier to the CPTAC dataset, but this will be very difficult with only 29 samples and some classes only included 2 samples. The other possibility is to perform exactly the same analysis including sample selection, gene selection, and consensus clustering of the CPTAC data independently to see whether the same "variable proteins" can be identified and the same subtypes can be discovered. The authors should also compare their proteomic subtypes with the ones previously reported by CPTAC.
6. To which extent proteomic subtypes/clusters reflect cell type composition difference?
7. Using two marker proteins as a guide to identify non-tumor proteins needs to be better justified. These markers were identified from transcriptomic studies, do they show good mRNA-protein correlation? The authors should also compare their quantification with other well-established deconvolution methods (e.g., ESTIMATE, Cibersort, etc).
8. Figure 2A, overlap between proteomic clusters and PAM subtypes needs to be statistically evaluated. Moreover, it is not clear how to define protein clusters for enrichment analysis based on the dendrogram. In addition, which genes (all genes, all protein coding genes, all quantifiable genes in the study, etc) were used as background for enrichment analysis (Gorilla and Hypergeometric testing)?
9. There are multiple gaps between novel peptides and their potential application as neoantigens. First, it is not clear whether they are expressed in normal tissues. Page 16, "CAGE and ribosomal profiling reveal they are transcribed and translated in other systems", what are the other systems? Were these peptides identified in any normal tissues? Secondly, MHC binding affinity of these peptides should be evaluated.
10. The authors tried very hard to justify that PAM50 genes have higher mRNA-protein correlations

even after variance adjustment. However, a very complicated method was used for the adjustment and the method was not formally evaluated, and it remains unclear why PAM50 genes tend to have higher mRNA-protein correlation if it is not because of higher mRNA variation.

11. The median mRNA-proteomics correlation (Figure 5A, 0.45) and median RPPA-proteomics correlation (Figure S1J, 0.47) are similar. Why RPPA measurements were considered as consistent with MS-based measurements (page 4)? If so, is this also true for mRNA measurements?

12. Association between CoTC 6 and glycolytic metabolites was used to support the added value of proteomics. This is interesting, but are any of the PAM50 subtypes associated with glycolytic metabolites? Did glycolytic enzymes/mRNAs show similar changes as the metabolites?

13. Figure 5C, the claim highly proliferative tumors had stronger mRNA-protein correlation needs to be tested statistically.

14. Proteomics data was quantified based on ratio to the tumor sample pool (equivalent to mean-centering). Was mRNA expression data also mean-centered before calculating tumor mRNA-protein correlation?

15. Figure 3C, overlaying protein cluster characteristics from Fig2A to demonstrate functional coherence of the subnetworks over-estimates true functional association. Enrichment analysis should be performed based on real gene annotations and the data should be provided.

16. The published metabolite analysis only quantified 18 metabolites?

17. Page 3, a reference is needed for the Oslo2 study

18. Page 6, the authors mentioned "The luminal tumor dominated CoTC groups (3, 4, and 6)". However, the CoTC group 6 seems to be dominated by Her2 rather than luminal.

Reviewer #3:

Remarks to the Author:

Manuscript summary:

The manuscript entitled "Breast cancer quantitative proteome and proteogenomic landscape" by Henrik J. Johansson et al. describes quantitative proteome and proteogenomic study of 45 breast cancer tumors across 5 subtypes defined by PAM50, including the commonly underrepresented normal-like subtype. The authors describe in-depth unsupervised classification based on whole-proteome data and compare the results with previous PAM50 mRNA-based classifications. The analysis enables the group to subtype tumors in more depth highlighting a novel subtype.

Additionally, the analysis includes protein-mRNA correlations highlighting that some of the PAM50 transcripts are good proxies for protein abundance while others are not. The study is further enhanced by proteogenomic analysis focusing on the protein expression of single amino acid variant peptides due to nsSNPs. Additionally, novel protein coding regions are identified in single cancer samples indicating potential for individual specific treatment regimes.

Overall, the manuscript reads well, and describes a landmark study of whole-proteome cancer stratification and results are validated using additional assays, but there are a few issues which should be addressed before publication can be considered.

Major revisions (arbitrary order):

In the proteogenomic analysis section of the manuscript and in the methods the authors describe the use of the human genome version GRCh37 and gene annotation GENCODE v19 with both being outdated versions since end of 2013. While the authors validate their potential novel findings with synthetic peptides, advancements in genome annotation are neglected (this includes the transition to GRCh38 and additional reassignment of non-coding genes to protein coding as highlighted in Wright et al., Nature Communications (2016) using the draft human proteome maps from Kim et al., Nature (2014) and Wilhelm et al., Nature (2014)). A quick scan of the claimed novel peptides revealed 10% of those now map to protein coding genes in Ensembl release 92 and 1% with partially deleted genomic sequence in the novel genome assembly. Additional 32% of novel peptides map to known sequences when accounting for an amino acid substitution. The authors should incorporate an additional step of filtering for novel peptide calls and check if the regions have been included in recent updates to the annotation as well as for additional SNPs in

the cancer samples.

Regarding the proteogenomic analysis: the authors use separate databases for their searches. This can lead to identification of different peptides for the same spectrum. The authors should incorporate a validation step and check overlap of spectra between their novel, SAAV, and reference peptide identifications and attribute these spectra to the already identified sequences.

The authors are rather stingy when it comes to citing other papers. For instance, several important papers on proteogenomics (incl. some published in Nature Communications) are missing. Similarly, proper references for other methods such as FASP or RPPA missing.

The introduction of the PAM50 panel misses the important aspect that it resulted in 5 different classification. The number '5' only suddenly shows up when the sample cohort is described.

In the results section for protein correlation analysis (page 5 and 10) the authors use subjective terminology such as clearly and robustly. Looking at the figures I do not agree with this assessment and would encourage the authors to justify their insights in a quantitative manner, e.g. using similarity measures and proper statistical evaluations between clusters. Similarly, the authors seem to miss to mention that the co-elevation of EGFR and MET appears to be confined to ductal carcinoma in situ regions FOR A SUBSET of normal like tumors. Figure 4E clearly shows that this is the case for 3 cases, but not for the 6 other cases. Please be more concise and evaluate the statistical significance of this observation.

Figure 5B: Since the different tumor types are spread over the entire range of proteome/transcriptome correlations, I don't think the authors' claim on page 12/13 (last/first paragraph) is justified. For instance, basal like and HER2 BC types are spread over the entire range. Please, tone down this statement and/or provide statistical support for this claim.

Figure 2: In the CoTC analysis, it seems that only 29 samples were considered. It is, however, not clear which 29 samples were used and which 16 samples were dropped and for which reason. Please explain.

Figure 2F: It is not clear which branch corresponds to which CoTC number – please provide this information.

The manuscript describes a true 'tour de force' which generated HUGE amounts of data. It seems that the authors try to show as much as possible in the figures. The result, unfortunately, is that many panels and their labeling is far too small to be useful. Thus, I strongly recommend to delete some of the panels and to enlarge the fonts to a legible size.

Minor revisions/typos (arbitrary order):

The authors alternate between terminologies using gene and proteins almost synonymously between the main manuscript and the supplementary figures. This leads to confusions specifically when it comes to the numbers of proteins/genes identified. Examples here is the identification of 13,997 protein products and 12,645 genes in the results section but the use of the term 'protein identifications' in Fig. S2 I when in fact 12645 genes are shown in the figure.

Abstract, line 4, reads proteome-based networks distinguishes BC subtype-specific functional protein modules; should be proteome-based networks distinguish

Page 16, paragraph 3; OSLO2 should be consistent with the remaining manuscript and read Oslo2

Figure 7 D: the three marked peptides belong to Inc-AKAP14-1:1 and not Inc-AKAP14-1:3

Page 5, reference to figure S2 (tumor composition correlated between MS and histopathology based evaluation (Fig S2G-I), should properly be Fig S2H-J.

Figure 4B: legend for the color codes is missing.

Figure 5B: 2 of the p-values are shifted to the right.

Reviewer comments

Reviewer 1:

1. I really liked the analysis linking the peptides to the non-coding genome and would have like to see some expanded analyses and focus on that aspect of the paper.

Were there specific peptides associated with ER/PR/HER2 status or the RNA/Protein classifications?

Thank you for the kind words of appreciation of our research!

The association of novel peptides to ER/PR/HER2 status or the RNA/Protein classifications is a really interesting question that we are working with. However, in our data set, detection of novel peptides is characterized by high level in mainly one tumor, which suggest it is a neoantigen characteristic. Many novel peptides are only identified in one quantification set, called TMT, which harbor quantification for 9 tumors and a reference composed of a pool of samples. The reference pool is used as denominator to calculate a relative ratio. The presence of samples in the pool makes it possible to identify peptides in a set of tumors which do not contain the protein product. See figure 7E for for examples of lncRNA protein products with tumor specific abundance. Hence, to conclude the relation of novel peptides to tumor subtype is difficult to draw based on this data.

To further study the tumor specificity of novel peptides we generated a MS proteome data set composed of 5 tumors with matched adjacent normal tissue. In this data we observe examples of both tumor-specific novel peptides and patient specific peptides (Fig S14H, I). Interestingly, peptides for lnc-AKAP14-1:3, which displayed high levels in a luminal A tumor among the 45 tumors in the BC landscape cohort also displayed high levels in another luminal A tumor in the new MS dataset with matched tumor and normal tissue (Fig 7E, S14H). Hence, the new data suggest that there is tumor specificity among the novel peptides but there might also be enrichment within subtypes. This is a question we aim to address further in future studies.

2. I also didn't understand the bars in Figure 7C. The legend referred to Figure S8 for details but I'm not sure how that helps. Do the black lines represent peptides found in the RNAseq and the MS from CPTAC?

Each vertical slice of Figure 7C represents a peptide in our data set linked to a 'non-coding' genomic region. The color-bar at the top of the figure specifies the type of genomic region a peptide is linked to. Each row containing black bars and white spaces indicates whether there is RNA-seq (middle bar) or peptide level (bottom bar) evidence from external publicly available data sets for the previously un-identified coding regions reported here by us. A black bar indicates evidence was found and a white space indicates no evidence was found. The caption has been modified in the revised manuscript and now reads:

Orthogonal evidence of novel peptides by public domain data, indicated by the presence of black bars in corresponding rows for RNA-seq (30), and re-analysis of proteomics data on breast tumors (5). See fig S14 for details.

3. Why were so few found in the MS data?

The reason for so few novel peptides were found the CPTAC data is probably related to the answer given in the first question about subtype specificity above. For the novel peptides we

find examples of tumor specific abundances which can lower the overlap. However, the presence of supporting RNA can suggest unknown translational regulation may allow for the presence of the transcript while preventing production of the protein in a patient specific manner. There is also a difference in analytical depth between the studies. This can be assessed by number of proteins with overlapping quantification across tumors or number of peptides used for identification and quantification. CPTAC dataset has a shallower proteome with proteins from 6405 genes quantified commonly across their tumors compared to our 9995. Comparing analytical depth using number of peptides used for identification and quantification is not possible since CPTAC do not show those numbers in their data. There is also the stochastic sampling of peptides in the mass spectrometer that can contribute to the discrepancy.

4. Why were the proteins restricted to the 9,995 genes that were present in all 45 tumors? There could be subtype-specific protein expression that might be incredibly useful for classification.

We agree that there might be useful information among the proteins which do not have quantitative values across all tumors. However, the MS technique used for quantification in this study can provide quantification across tumors regardless of subtype specific protein expression. If there is an identification and quantification for a peptide, all tumors in the TMT set will be quantified. There will always be a background signal even if the protein is absent in the subtype. See question below for details about TMT quantification.

However, using the proteins without quantification is technically challenging since it is to our knowledge not possible to impute protein quantification between tumors. There is also the presence of plasma, erythrocytes, adipocyte and immune cells that should be imputed. The overlap in quantification between tumors will be different for different proteins, making analysis challenging and comparisons of findings between tumors and between proteins problematic. To our knowledge, there is no good way how to handle this problem and since we quantify proteins from 9995 genes across the tumors compared to 6405 for the 2nd deepest study (Mertins 2016 Nature) we used this depth for all data analysis.

5. I looked at the centroid correlations for your 45 tumors and while the Basal--like and Luminal A are very strongly those subtypes, the selections of tumors for the other subtypes are not really that strong as shown in Table S1 and Figure S2A. Some of the correlations for the HER2 samples were really low and for the normal--like there were a few that were equally high for both normal-like and Luminal A. It seems a bit unfair to say that RNA classification was poorer when these were not really prototypical samples to start with.

Tumor samples were selected randomly (within each subtype) for inclusion in this study to be representative of the PAM50 subtypes and their known ambiguities. The distributions of correlations to each of the 5 PAM50 subtypes in the 45 tumors represent that among the 373 tumors in the Oslo2 cohort (Fig R1 below). Note that normal-likes often have high correlation also to luminal A and that HER2 subtype often also have high correlation to luminal B. The PAM50 subtype is assigned to the centroid with the highest correlation. In some cases, the correlation can be relatively small, and there can be a small difference between the best correlation and the second best. Hence, based on this method of classification, some tumors will be borderline between subtypes and some tumors are more characteristic of a subtype than others.

Fig R1. Correlation of mRNA to PAM50 centroids for PAM50 assignments for 373 tumors in the Oslo2 cohort.

We look to enhance current BC classifications to address their known shortcomings, including uncertainties in classifying HER2 and normal-like tumors. To clarify, we do not claim transcript-based classifications are poorer than proteome-based classifications, we claim there is information within the proteome that can be used in clinical situations when current molecular (transcript-based) classifications are inconclusive. We could not use characterization of the proteome to address uncertainties in diagnoses if tumors were selected to be unambiguous representations of clinical subtypes.

6. CoTC1 and CoTC2 wouldn't separate in the PAM50 analysis because the set of genes was enriched for tumor- intrinsic genes versus non-tumor genes such as the immune proteins that separate those groups.

Precisely! Thus the body's systemic response to the tumor may be valuable for classification and treatment stratification.

7. Based on the CoTC dendrogram visualization in Figure 2C, it seems like group 6 should have been separated into 2--3 groups.

We initially had the same notion that CoTC6 would split into 2 groups (Fig 2C, S4I). To test this we performed consensus clustering and allowed for more than the 6 groups suggested by the delta plot for cumulative distribution function (S4H) to be formed. However, upon increasing the number of clusters, the tumors in CoTC6 still remained together while other groups started to break up. This suggests that overall the tumors in CoTC6 are similar but there are some proteins that separate them. We are currently assembling a new set of tumors with focus on the HER2 and luminal B subgroups for the potential to define new groups based on protein levels with more tumors.

The dendrogram distances in fig 2C, D are related to weight, but not directly proportional, and should be used qualitatively for visualization purpose. Figure S4I gives a better quantitative view of the distance between subtypes.

8. Is there concern with the limited sample numbers after reducing to the subset of samples, that the clustering is more driven by individual differences.

All reviewers have raised questions in one way or another about the number of tumors used for clustering in the study. Thus we provide a uniform answer covering all questions. We apologize if the answer to your specific question does not appear at once.

We are aware that the sample size is relatively small, especially when considering the number of samples in genomics and transcriptomics breast cancer studies today. Even though mass spectrometry based proteomics technologies have matured a lot during the recent years there is still a limitation when it comes to number of samples for analysis in relation to analytical depth. The compromise stands between more samples and less analytical depth or fewer samples and larger analytical depth. With analytical depth we here mean the number of spectra that are acquired for each sample and corresponds to number of unique peptides that are used for protein identification and calculation of protein levels. Acquiring more spectra per protein for each sample increase the number of proteins identified and the confidence in the protein identification and the quantification accuracy. To increase the number of acquired spectra, both per protein and to reach low abundant proteins, peptides are fractionated before nanoLC-MS/MS to increase analytical depth. Increasing peptide fractionation before nanoLC-MS/MS increases instrument analysis time and decreases sample throughput. The stochastic nature of MS acquisition, which in part depends on different sample compositions, also means that different peptides and hence proteins will be identified in different samples, decreasing the number of proteins with quantitative overlap that can be used for analysis as the number of samples increase. Increased fractionation and concomitant acquisition of multiple peptides (increased analytical depth) from the same protein preserves the overlap in quantification better between samples. We have previously developed a method to acquire deep proteome coverage to quantify proteomes and perform proteogenomics (Branca 2014 Nature Methods, Zhu 2018 Nature Comm) and used this method to acquire, as far as we are aware of, the deepest proteome coverage across a tumor cohort to date. The only other study to profile the proteome of breast tumors anywhere near as deeply as we have includes 77 tumors after quality control (Mertins et al., 2016 Nature). Our study does include fewer tumors, however we are the first to recapitulate known subtyping with an unbiased analysis of the whole measured proteome. This finding that breast cancer biology is driven by the proteome is fundamental, and recapitulating known biological subclasses provides evidence that the 45 tumors are adequate for subtype analysis based on high quality proteomes. The hallmark study by Botstein and colleagues (Nature 2000) classified 42 tumors into molecular subtypes based on their mRNA signatures provided the basis for the concept of breast cancer subtypes today. Hence, a relatively small cohort size can grasp the major molecular themes in breast cancer. The RNA based subtypes have been methodically characterized in a multitude of datasets, and the proteome subtypes need to be confirmed in additional cohorts as the field moves forward.

The details of how the consensus core tumor clusters (CoTC) were generated is given in the materials and methods section, starting with a summary, as shown below:

Defining proteome based tumor subtypes - CoTCs

Cluster Detection - overview

In order to identify the strongest subtypes in the protein dataset, we first identified and filtered out unmodulated genes (genes with low standard deviation). From this list we then removed genes that could be associated to adipose, plasma, erythrocytes and immune pathways from the lists defined above, to obtain as pure list of tumor proteins as possible. Next we identified and ignored all outlier samples that could weaken the cluster generation.

The remaining samples / genes were then subjected to consensus clustering in order to identify consistent subtypes in the dataset. All code can be found in “Generate_Clustering_and_Network.R” script.

Based on the knowledge from the correlation of mRNA to the different PAM50 centroids (see Fig R1 and S2D) some tumors have dual and borderline characteristics between subtypes. This could also be seen at the protein level where some tumors mix with other subtypes, or present weak association to a cluster, when using all identified proteins or the PAM50 set of proteins (Fig 2A, B). Performing clustering with tumors that are borderline between subtypes will result in weak and undefined clusters. To mitigate this we demanded that tumors included in the clustering have >0.5 correlation to at least one more tumor (Fig S4F). This inevitably shrunk the number of included tumors in the clustering analysis, but based on subsequent enrichment and protein abundance analysis of the groups suggest the major characteristics of the PAM50 are captured as expected, and adding additional information from immune infiltration and extracellular matrix (Fig 2E, 3D).

To confirm the validity of the CoTCs groups with only 2 members (CoTC2 and 4) we first identified protein groups that are characteristic of the groups. Both CoTC groups are immune enriched, CoTC2 for the whole immune cluster and CoTC4 for interferon alpha response (Fig 2A, E, 3D, S2), and both showed good mRNA-protein correlation (Fig S5C, R2A). Since CoTC2 is composed of basal-like tumors and CoTC4 of luminal B tumors we selected out these tumor subtypes from the whole Oslo2 cohort. Clustering of the immune RNA group for CoTC2 in basal-like and interferon alpha response RNA group for CoTC4 in luminal B tumors revealed the presence of additional tumors with similar expression in the whole Oslo2 cohort (Fig S5D, E, R2B, C).

Figure R2. Presence of CoTC2 and CoTC4 like tumors in Oslo2 cohort. **(A)** mRNA-protein correlation for non-tumor components, adipose, erythrocyte, plasma and immune, defined in fig S3. Immune and interferon alpha response being characteristic of CoTC2 and CoTC4, respectively (Fig 2A, E, 3D, S2) **(B)** Oslo2 basal-like mRNA expression of immune defined genes from the proteomics data (Fig S3), characteristic of CoTC2 (Fig 2A, E, 3D, S2). **(C)** Oslo2 luminal B mRNA expression for interferon alpha response genes enriched in CoTC2 (Fig 2 E, 3D).

We see the CoTC subtypes as an indication that current classifiers are incomplete and provide a suggestion as to how they could be improved. The emergence of CoTC2 suggests therapy assignments for patients diagnosed with rapidly proliferating basal-like tumors may be improved by characterization of immune component infiltration.

Cohort size is always a concern and this study was designed to include 45 tumors to strike a balance between depth of profiling and breadth of inclusion. We believe the cohort of 45 tumors to be adequate for subtyping based on high quality quantified proteomes because unbiased clustering of protein abundance profiles stratifies tumors based on breast cancer biology.

9. The CPTAC data looks more like 3 groups than 6.

The CPTAC breast cancer dataset was used with the intention to provide a view of the CoTCs in an independent breast cancer cohort. However, the CPTAC dataset is different to ours in a number of ways. Firstly, it only defines 4 PAM50 subtypes (Basal-like, HER2, luminal B and luminal A), and lack the normal-like subtype. Secondly, their original clustering of the data resulted in 3 clusters where basal-like tumors intermix with other subtypes. The basal-like subtype known to be very different from luminal tumors (Hoadley 2014 Cell). Thirdly, there has been quality issues with the CPTAC dataset, which is mentioned in their paper and via personal communication the authors. Fourthly, the CPTAC dataset has a shallower proteome with proteins from 6405 genes quantified commonly across their tumors compared to our 9995. For the proteins defined as highly variable across tumors and used for clustering in our dataset (fig 2C, S4), half of them were identified in the CPTAC dataset and used for clustering (fig S5A, B). Fifthly, the dynamic range for proteins with known differences between PAM50 subtypes is larger in the Oslo2 landscape cohort compared to CPTAC MS data (Fig R3). The improved dynamic range in our study is likely due to our extensive fractionation that decrease co-isolation of neighboring precursors, and thus increase the TMT dynamic range.

Figure R3. Examples of proteins with known differences between PAM50 subtypes in Oslo2 landscape cohort and CPTAC MS proteome data.

Despite these discrepancies between the datasets, we used this dataset since it is the only dataset available for a repeated observation at the moment. We observe three major clusters in the CPTAC dataset using our defined list of proteins, corresponding to a basal-like, a luminal A and a mixed luminal-B and HER2 cluster. Hence, to some degree resembling three of the proteome clusters in our dataset, CoTC1 (basal-like), CoTC3 (luminal A) and CoTC6 (mix of luminal B and HER2).

10. For figure 4E and F, would it be better to have the image show the correlation between MET and EGFR expression than the correlation for each with its respective RNA since you already showed they were similar? It might help with the interpretation of 4E and F.

We assume that the question relates to fig 4D. Good suggestion, instead of showing box plots for each PAM50 subtypes the figure has now been exchanged to scatter plots showing how the RNA values for MET and EGFR correlate in the whole Oslo2 and TCGA cohort. The new plots adds the information that EGFR and MET are high in the same tumors for basal-like and normal-like subtypes (Figure 4D).

11. In Figure 5B, is the lower correlation with ribosome and spliceosome more a function of difficulty in mapping RNA transcripts due to either highly similar genes (ribosomal) or smaller genes (spliceosome) and therefore the RNA quantification is not as accurate in those genes?

The variance of ribosomal and spliceosomal mRNA (and protein) is generally small (Fig S11K-N). We observe a significant statistical difference in mRNA variance between ribosomal and all genes in our dataset (F-test, $p < 2.2e-16$). The same is true at the protein level, suggesting that the true biological variation between samples is small both on mRNA and protein levels for these genes. This furthermore suggests that the biological variation-to-technical variation ratio is generally small for these mRNA and protein measurements. This means that even if mRNA and protein are highly correlated biologically we will not be able to observe this correlation from our measurements. For the clinical mRNA markers (signatures) a certain level of variance in mRNA expression must be present. In general, high variance in mRNA expression does not imply high correlation with protein. What we have shown, however, is that for the clinical mRNA markers we also have high correlation to protein. This establishes a clear link between mRNA markers and protein phenotype.

12. For figure 5C, is the poor tumor-protein correlation at all linked to tumor purity levels? The samples with the lower overall correlation also had higher non-tumor proteins expressed. If the RNA and Protein were from separate pieces of tumor, the cellular heterogeneity could also account for the poorer correlation. How were the proteins selected to show in 5C?

We received questions related to figure 5C from all reviewers. Hence, we perceive that it was difficult to follow and made a new figure, which can hopefully more clearly convey the message we wanted to bring forth.

However, first we want to clarify the old figure 5C. The intention with figure 5C was to illustrate the differences in mRNA-protein correlation between tumors. The mRNA-protein correlation for each tumor is shown in figure S6L (now S11 in the reworked version of the manuscript). We ranked the tumors according to their mRNA-protein correlation and associated groups of proteins to tumors with high and low mRNA-protein correlation. Protein quantification were visualized in old figure 5C. Proteins with a positive correlation are significantly enriched in proteins related to transcription, splicing, translation and cell cycle. Proteins with a negative correlation is enriched in extracellular region related proteins. For the new figure, as in the old one, we associated proteins to the tumor mRNA-protein correlation by Spearman correlation. In an attempt to clearly associate statistical support to the analysis, we performed ranked gene set enrichment using all correlated proteins, and

visualized significantly enriched protein groups (all with p-value <1E-17) in the new figure (Fig 5D). Terms positively associated with tumor mRNA-protein correlation are related to proliferation, transcription, splicing and translation, while extracellular and plasma related terms are negatively associated.

The sentence: Highly proliferative tumors (basal-like, HER2, luminal B, and/or high MKI67) appear to have more strongly correlated proteomes and transcriptomes (Fig 5C and S6L).

Has been toned down to:

Highly proliferative tumors (basal-like, HER2, luminal B, and/or high MKI67) have a tendency to have more correlated proteomes and transcriptomes than lowly proliferative tumors (luminal A, normal-like) (Fig S12A).

To assess if the poor tumor-protein correlation are at all linked to tumor purity levels we compared the tumor mRNA-protein correlations to histopathology. Histopathology assessment of tumor compositions displayed a weak, although positive correlation while stroma displayed a weak negative correlation (Fig R4). Negative association of stroma to tumor mRNA-protein correlation is in line with the association study in figure 5D.

Figure R4. Association between histopathology assessment of tumor composition and tumor mRNA-protein correlation.

We agree that since RNA and Protein were from separate pieces of tumor, the cellular heterogeneity could also account for some of the poor correlations. The protein, RNA and histopathology measurements were from separate neighboring pieces, which can explain some of the poor correlations. However, the overall trend among the comparison of tumor mRNA-protein correlation to protein groups and histopathology suggest residual characteristics.

Minor comments:

13. Figure 3A says correlation between protein and RNA, but you aren't showing correlation. Just the plots of the RNA and Protein separately.

The intention with the figure was to show how well the quantitative values for a protein complex correlates. But it is correct that we don't show correlation per se.

The figure text now reads:

Protein and RNA levels across tumors for known protein complexes (Fig S4A, B examples of more complexes).

14. Figure S1, what is a TMT10 set?

When performing quantitative mass spectrometry based proteomics we use a method denoted as isobaric tags. TMT is one of the isobaric techniques available. For quantitative purposes, tryptic peptides in a sample is covalently labeled with a mass reporter and a mass balancing group. Another sample is labeled with a different reporter mass and a balancing group to yield the same overall mass attached to each peptide as for the other mass reporter. For TMT10, there are 10 different reporter masses, and so allows analysis of multiple samples at the same time. To analyze more than 10 tumor samples we connect the different TMT10 sets by using one of the mass tags in each set as a reference. The reference samples contain a pool of all samples to be representative of the diversity in the samples. All other samples in the TMT set is related to the pool to generate a ratio. Robust protein quantification is dependent on the number of peptides used for protein quantification. To provide an understanding of the datasets good quantitative robustness in each set and across TMT sets we provide the plots in Fig S1.

15. Figure S8, what is the ranking of the peptides? Also this figure has a legend for K, but only goes to J.

For visualization purposes the peptides are ranked from highest value to lowest value (left to right) in the plots.

We are sorry for the additional legend. The legend for is a remnant of a previous version of the figure and is now removed.

Reviewer 2:

First, the key contribution of the study is not clear.

The aim with the study is to provide a broad proteome centered multi-omics analysis of breast cancer which can provide a resource for further analysis. To attain this aim we initially need to establish that the data levels reproduce known breast cancer biology and analyze what knowledge can be attained by the in-depth proteome molecular phenotype analysis.

To make it clearer, we are listing the key contributions below:

1. The data set itself including mRNA, protein, phosphoprotein, copy numbers, SNV data and metabolite abundances across 45 breast tumors is a key resource contribution to the field of breast cancer research.
 - a. This data set includes the deepest profiling of the breast cancer proteome to date, with proteins from 9995 genes quantified commonly across all tumors compared to 6405 for the 2nd highest by Mertins et al.
 - b. It is the first BC multi-level dataset to harbor metabolomics data
 - c. The data set is the first large proteome dataset to contain all the 5 PAM50 subtypes. The previous ones having neglected the normal-like subtype.
 - d. We are also first to provide an easily queryable data portal, including the breast cancer proteome, with analysis tools to ensure that this rich dataset can be explored by the research community.
2. Our deep proteome data is the first to reproduce known breast tumor subtyping by unbiased clustering. This is fundamental knowledge, building on the central dogma, which the previous proteomics papers (Mertins 2016 Nature, Tyanova 2016 Nature Comm) have not been able to reproduce.
3. Tyrosine kinase receptor ligands in the plasma component of tumors are correlated to phosphorylation status of oncogenes. This association highlights that the role of plasma in tumorigenesis may be underestimated since the majority of BC studies up to date have relied on RNA measurements which lack plasma information. Only proteomics methods can characterize the plasma protein composition and tumor phosphorylation status.
4. Proteome-based correlation network distinguishes protein modules relating to breast cancer biology and breast cancer subtype expression. This finding highlights key protein components in breast cancer.
5. A subgroup of luminal B and HER2 tumors (all part of the CoTC6) show direct evidence of the Warburg effect by depleted glucose and elevated lactate/alanine levels. This is the first time, to our knowledge, that the high proliferative tumors of luminal B and HER2 have been able to be linked to the Warburg effect.
6. Protein co-expression analysis of FDA approved drug targets reveal highly correlated clusters, suggesting correlated drug targets work in concert and may be promising co-targets. As example of potential drug target combinations, we chose two known oncogenes, EGFR and MET, with high RNA and protein levels in basal-like and normal-like tumors for IHC validation of EGFR and MET in our dataset and an independent cohort of 530 tumors. This is the first time that the normal-like subtype is associated with high levels of EGFR and MET, with the possibility to better define this understudied tumor group.
7. Correlation of mRNA and protein is higher in gene expression signatures than expected by chance. This suggests that the genes in the gene signatures have been

selected based on them heeding the central dogma. However, 30% of the genes in the dataset lack significant mRNA-protein correlation which means that protein measurements is the only means to grasp their phenotype. To emphasize the importance of the mRNA-protein correlation analysis, we have added figure 5C which show mRNA-protein correlation for genes causally associated with cancer from COSMIC and breast cancer (Nik-Zainal 2016 Nature). These causally associated genes display varied mRNA-protein correlations, indicating that some genes can not be studied by RNA as a surrogate measurement and require protein measurements (Fig 5A-C).

8. A minor contribution since it is a repeated observation (2nd cohort) is the finding that copy number alterations are dampened at the protein-level, which in part is mediated by the ubiquitin system. We perceive this as important additional data since breast cancer is largely considered a copy number driven disease (Curtis 2012 Nature).
9. Application of our recently developed IPAW workflow (Zhu et al., 2018 Nature Communications) identifies peptides derived from genomic regions thought to be non-coding or corresponding to alternative translations of known genes (Fig 7B). This is the first such report regarding breast tumor tissue and suggests proteogenomic analyses can be applied to discover immunotherapeutic targets by identifying peptides derived from ncRNAs, pseudogenes, introns, and intergenic regions. Of note, Mertins et al identified single amino acid variant peptides and novel splice junction peptides only from their proteogenomics workflow.
10. The proteogenomic analyses reveal that single amino acid variants (SAAVs) impact protein levels in breast tumors. This has not been shown in breast tumors before. SAAV peptides corresponding to homozygous and heterozygous SNPs can result in protein variant-specific level changes. The SAAV data also adds to the catalogue of variants that actually reach the proteome.

We believe a broad approach illustrating the utility of the whole data set is appropriate for this study because the data itself and its wide utility is our focus. Our perception is that the points above are mentioned in the manuscript and extending the results and discussion parts to put more emphasis on each of the points above will make the paper too long and difficult to read.

Second, the sample size is very small, especially for subtype analysis

Please see the combined answer with reviewer one's question:

8. Is there concern with the limited sample numbers after reducing to the subset of samples, that the clustering is more driven by individual differences?

Third, some quantitative statements are not supported by statistical analysis.

This comment is addressed in the responses to the specific points made by the reviewer below.

1. This study is very similar to the CPTAC breast cancer study (Nature, 2016), and most of the findings have been previously reported in this and other proteogenomic studies

The concept of collecting and integrating proteomic and transcriptomic data sets to study breast tumors is common to both our and the CPTAC breast cancer studies. However, our findings are fundamentally different and sometimes conflict with those of the CPTAC study (Mertins, et al. 2016). For instance, Mertins et al. largely conclude that an unbiased analysis of the proteome yields tumor stratifications differing from mRNA-based classifications while we find unbiased analyses of the proteome is consistent with and refines these subclasses. Additionally, we find the co-expression of EGFR and MET to distinguish a subclass DCIS in normal-like tumors (supported with high resolution immunohistochemical imaging). We delineate aggressive luminal B tumors based on immune component infiltration. We find analysis of metabolite abundances to stratify tumors based on their known growth phenotypes. We find phosphorylation status of specific BC driver proteins to be linked to specific protein-class abundances. We identify gene expression signature to have higher than expected mRNA-protein correlation. We identify potentially tumor-specific peptides derived from previously believed non-coding regions of the genome. All of the above has not been reported previously. Please also see answer to key contributions above for a complete list of novel findings.

1.1 proteomic subtypes overlap with transcriptomic subtypes but provide novel information has been reported in other proteogenomic studies

The key distinction in our study is that unbiased analyses considering the whole measured proteome largely reproduce known subtypes that are based on breast cancer biology. We are the first to report this. There is a fundamental difference between selecting proteins (i.e. PAM50 genes) to classify tumors into known subtypes (has been done) and using the entire measured proteome or the most varying to do so. This data allows the proteome-wide analysis in relation to RNA-level findings.

1.2 impact of DNA copy number alterations is dampened at the protein level has been reported in other proteogenomic studies

Yes, this is correct, dampening of copy number effects at the protein level has been reported before in the CPTAC BC dataset. However, breast cancer is considered to be largely driven by copy number changes (Curtis 2012 Nature), and we find it important that we can contribute to the reproducibility of science and substantiate the finding that the majority of copy number changes are dampened at the protein level, in an independent breast cancer cohort. To us, this radically changes the way how a potential copy number driver should be assessed. Hence, copy number changes that reach the proteome can serve as guide for future exploration and selection of copy number drivers in BC.

We also provide additional analyses to support that ubiquitylation and proteasome degradation are mechanisms for dampening DNA copy number alteration affects at the protein level.

1.3 identification of novel and SAAV peptides has been reported in other proteogenomic studies

That is correct, by us and others, we do not claim this is the first attempt to identify novel and SAAV peptides.

SAAV variations play a major role across nearly every aspect of cellular function and disease and we believe expanding the list of known variants that reach the proteome is important. Just because sequencing efforts, and there are multiple of them published in high impact journals (e.g. Curtis 2012 Nature, Ellis 2012 Nature, TCGA 2012 Nature, Shah 2012 Nature, Stephens 2012 Nature, Nik-Zainal 2012 Cell, Nik-Zainal 2016 Nature, Pereira 2016 Nature Comm, Shlien 2016 Cell Reports) have reported a mutation of non-synonymous SNP does not mean it will be reach the intended protein product. Expanding the compendium of SAAV peptides is important to gain understanding of impact of the underlying mutation or non-synonymous SNP.

We have applied an in-house tool SpectrumAI (Zhu Y. et al *Nature communications* 2018) to retrieve high confidence identifications of SAAVs previously un-reported from high quality proteomic measurements and shown that the presence of these variants impacts protein abundance in the breast tumor cohort. This has not been shown before in breast cancer. Providing high-quality SAAV identifications are also essential for evaluation as potential neoantigens. We are also first to report alle specific protein levels.

Regarding the novel peptides, we are first to show quantitative analysis of novel peptidome (peptides detected from previously undescribed protein-coding genomic regions) in the breast tumors and illustrate a few examples of elevated expression of translation products from lncRNAs in individual breast tumors.

Mertins et al proteogenomics analysis identified single amino acid variant peptides and novel splice junction peptides. In addition to these peptides we also searched for and identified peptides from previously thought to be non-coding RNA, alternative translation sites, intergenic regions and pseudogenes (Fig 7B). This is to our knowledge, the first time this analysis has been done in breast cancer. Providing a compendium of novel peptides is also important for developing immunotherapy based on neoantigens.

1.4 There are some potentially interesting findings, but they are mixed with other results, and thus the major contribution of the manuscript is not clear. It would be very useful to separate new findings from results that simply confirm previous findings.

We have chosen to organize this manuscript by analysis. For each analysis we show consistency with what is known or has already been reported to provide confidence in the quality of our data before continuing to make novel observations, and in several cases validating those observations. Confirming consistency with prior findings, when applicable, was a necessary step before building on them. In each section we clearly state and reference when observations are consistent with prior reports. If no references are provided, originality of the finding is implied. We decided to organize the manuscript in this manner to maintain a continuous focus on each aspect of the data. Thus, our thought processes can be followed from beginning to end without interruption. If we were to list the consistencies with prior reports and novel observations separately, we would need to go over each aspect of the data twice and be quite repetitive in introducing each concept.

1.5 The manuscript spends too much space on the latter [confirming previous findings], and the potentially interesting stories are not fully developed.

We aim to provide a high-quality dataset with examples of how hypotheses can be explored by integrating the different systems levels (see also answer to key contributions above). We develop hypotheses across all levels of the data pursuant to our own specialties and present

novel translational findings summarized in our Abstract and Discussion. A major aim of the current work (embodied by the accompanying tool which will be available to the public) is to encourage researchers with expertise outside of our own that this dataset may be useful to direct their inquiries.

This said, we see the point by the reviewer and have gone through the manuscript, and for example moved the evaluation of clustering in CPTAC data (fig 2E) to supplementary to give more space and emphasis to novel findings.

1.6 For example, it would be nice to have experimental data to evaluate the functional impact of co-targeting MET and EGFR.

We agree that it would be nice to have this experimental data. This is a hypothesis that stood out from an analysis of protein abundance correlations of FDA approved drug targets. Unfortunately, there is a lack of cell line models for normal-like tumors and developing this model is beyond the scope of this article. However, Supporting our finding in normal-like cancer, for triple negative breast cancer cell lines and PDX models, combined EGFR and MET targeting have been shown to be more efficient than monotherapy (Yi 2015 Int J Oncology, Linklater 2016 Oncotarget).

This said, we feel that the normal-like subtype is understudied and lacks attention from the research community, and we will continue to look for research models to investigate it. With the new data showing high levels of EGFR and MET in normal-like tumors we also hope to foster novel investigation from the breast cancer community.

2. Multiple methods were used to derive sample clusters based on proteomics data, including clustering based on all genes, clustering based on PAM50 genes, subtyping/consensus clustering based on selected genes and samples. This is very confusing.

Treatment assignments for breast cancer patients are largely (if not solely) based on subtyping tumors by various means (IHC, PAM50, Mammaprint, OncotypeDX etc.). Thus research efforts to improve treatment assignments and identify biomarkers must explore a different way of subtyping. Clustering based on all genes illustrates the information currently used to stratify patients into treatment courses is intrinsically contained within the proteome (a fundamental finding), subtyping based on smaller sets of genes (such as the PAM50 set) illustrates that the same information can be garnered from fewer and thus more clinically tractable measurements, while defining the consensus clusters based on highly variable proteins allowed for refining current subtypes based on properties such as immune component infiltration. All subtyping methods are specified in figure captions and properly referenced in the text.

2.1 It is not clear which classification best represents proteomic heterogeneity.

We did not attempt to optimize the representation of proteomic heterogeneity, rather to explore how the proteome can better assign patients to treatment regimens or provide new fundamental insights into breast cancer biology. As mentioned in the answer to the question above, all applied classification methods made unique contributions.

3. *Only 29 samples were used for subtyping/consensus clustering. The sample size is very small for subtype analysis. Moreover, these samples were divided into 6 classes (CoTCs), and most subtypes included only a few samples. The authors highlighted CoTC2 as a new finding, but it only included 2 samples. A subtype with only 2 samples is very questionable.*

Please see the combined answer with reviewer one's question:

8. *Is there concern with the limited sample numbers after reducing to the subset of samples, that the clustering is more driven by individual differences?*

4. *The authors tried to validate their proteomic subtypes in the CPTAC dataset and claimed that "Similar results were obtained when applying the unsupervised clustering on CPTAC MS-based quantified breast tumor proteomes". However, this was not supported by any quantitative evaluation. Actually, Figure S3I and S3J look very different to me. For example, Luminal A and Basal samples are perfectly clustered in S3I but not S3J.*

The CPTAC breast cancer dataset was used with the intention to compare the datasets. However, there can not be a direct comparison since the CPTAC dataset is different to ours in a number of ways. Firstly, it only defines 4 PAM50 subtypes (Basal-like, HER2, luminal B and luminal A), and lack the normal-like subtype. Secondly, their original clustering of the data resulted in 3 clusters where basal-like tumors intermix with other subtypes. The basal-like subtype known to be very different from luminal tumors (Hoadley 2014 Cell). Thirdly, there has been quality issues with the CPTAC dataset, which is mentioned in their paper and via personal communication the authors. Fourthly, the CPTAC dataset has a shallower proteome with proteins from 6405 genes quantified commonly across their tumors compared to our 9995. For the proteins defined as highly variable across tumors and used for clustering in our dataset (fig 2C, S4), half of them were identified in the CPTAC dataset and used for clustering (fig S5A, B). Fifthly, the dynamic range for proteins with known differences between PAM50 subtypes is larger in the Oslo2 landscape cohort compared to CPTAC MS data (Fig R3). The improved dynamic range in our study is likely due to our extensive fractionation that decrease co-isolation of neighboring precursors, and thus increase the TMT dynamic range.

Figure R3. Examples of proteins with known differences between PAM50 subtypes in Oslo2 landscape cohort and CPTAC MS proteome data.

Despite these discrepancies between the datasets, three major clusters can be observed in the CPTAC dataset using our defined list of proteins, corresponding to a basal-like, a luminal A and a mixed luminal-B and HER2 cluster. Hence, to a degree validating three of the proteome clusters in our dataset, CoTC1 (basal-like), CoTC3 (luminal A) and CoTC6 (mix of luminal B and HER2).

We are grateful for the observation by the reviewer about the miss-statement of the sentence: Similar results were obtained when applying the unsupervised clustering on CPTAC MS-based quantified breast tumor proteomes considering the same subset of proteins limited to those quantified across all samples that the authors deemed high quality (632 of 1334) (Fig 2E, S3J).

The sentence have now been changed it to:

Unsupervised clustering of CPTAC breast tumor proteomes ⁵, using the overlapping high variance proteins (632 of 1334), identifies three tumor clusters which resemble CoTC1 (basal-like), CoTC3 (luminal A) and CoTC6 (mix of luminal B and HER2) (Fig S5A, B).

5. In order to formally use CPTAC data as a validation set, one possibility is to come up with a classifier from the current dataset and then apply the classifier to the CPTAC dataset, but this will be very difficult with only 29 samples and some classes only included 2 samples. The other possibility is to perform exactly the same analysis including sample selection, gene selection, and consensus clustering of the CPTAC data independently to see whether the same “variable proteins” can be identified and the same subtypes can be discovered. The authors should also compare their proteomic subtypes with the ones previously reported by CPTAC.

We were not considering to use the CPTAC as a validation data set since it is quite different from ours. See answer to previous question. Building and testing a classifier with these differences between the datasets will provide inconclusive data which will be difficult to interpret. Instead we wanted to provide a glimpse of what information our set of highly variable could provide in another dataset.

Mertins et al applied standard deviation in the same way as we did to select variable proteins for consensus clustering. Differences comes to cutoffs, where Mertins et al used a fixed cutoff of 1.5 SD to select 1521 in their dataset, while we used a gaussian mixture model to select 1334 variable proteins in our dataset. Both Mertins and us performed consensus clustering to define 3 and 6 clusters, respectively.

Comparing the proteomic subtypes with the ones previously reported by CPTAC is a great idea. We have now added Mertins cluster annotations to the clustering of CPTAC data using the overlapping most varying proteins in our dataset (Fig S5B). However, we have not generated a supervised classifier due to differences (described above) between the two datasets. It is not possible, since a number of our discriminates on protein level are not detected cross CPTAC dataset.

6. To which extent proteomic subtypes/clusters reflect cell type composition difference?

This is an interesting question. From the heatmap in figure 2A with clustering using all 9995 proteins with quantification across all tumors we deduce that part of the basal-like cluster is driven by immune components. Also, plasma components are higher in normal-like samples (figure 2A). For the CoTCs, the enrichment analysis show that CoTC2 and CoTC4 is enriched in immune related proteins (Figure 2E). The immune contribution to CoTC2 and CoTC4 can also be observed when overlaying the quantitative values for each CoTC onto the protein correlation network (Figure 3D). To extend on this analysis, we have added to the manuscript, the percentage of non-tumor components of each sample (Figure S2E, S4K). Non-tumor definition is described in the answer to the review question below and figure S3. The tumors in the basal-like cluster in figure 2A with immune components and CoTC2, both show higher percentage of immune defined proteins compared to other tumors. CoTC4 display similar immune percentages as the other tumors, which could be due a different (smaller) set of immune cells as indicated by the enrichment analysis (Fig 2E).

To investigate what is driving the clusters, we tested inter-variability between tumor subgroups (in CoTC and proteome clusters (from Fig 2A)) by Kruskal-Wallis test (Figure S6). The differences between subtypes (CoTC and proteome clusters (from Fig 2A) is mainly attributed to proliferation, estrogen and immune as part of the most significant terms (Fig S6).

7. Using two marker proteins as a guide to identify non-tumor proteins needs to be better justified. These markers were identified from transcriptomic studies, do they show good mRNA-protein correlation? The authors should also compare their quantification with other well-established deconvolution methods (e.g., ESTIMATE, Cibersort, etc).

Good point! In our initial draft of the paper we had a detailed supplementary figure outlining how the non-tumor proteins were defined. See new supplementary figure S3. Concerning the markers, CD5 and PTPRCAP are know not just from Uhléns paper (Uhlen 2015 Science) as immune markers. CD5 is a pathology marker used to identify T-cells while PTPRCAP associate with tyrosine phosphatase CD45, a key regulator of T- and B-lymphocyte

activation. The adipocyte marker FABP4 is a fatty acid binding protein found in adipocytes and PLIN1 coats lipid storage droplets in adipocytes. The erythrocyte marker HBB is self-explanatory while SPTA1 increase erythrocyte plasma membrane elasticity and deformability. ALB and A2M are also well known plasma proteins. The mRNA-protein correlation for the immune and adipocyte markers are all significant. The mRNA-protein correlation for the defined non-tumor groups can be seen in fig S5C. As expected, erythrocyte and plasma proteins have low mRNA-protein correlation compared to immune and adipocytes. The immune defined group displayed good correlation to the two enrichment tools, ESTIMATE and xCell, and a deconvolution algorithm, CIBERSORT (Fig S3G). Interestingly, the MS based immune list showed the highest correlation to histopathology based assignment of immune components.

8. Figure 2A, overlap between proteomic clusters and PAM subtypes needs to be statistically evaluated. Moreover, it is not clear how to define protein clusters for enrichment analysis based on the dendrogram. In addition, which genes (all genes, all protein coding genes, all quantifiable genes in the study, etc) were used as background for enrichment analysis (Gorilla and Hypergeometric testing)?

We have now evaluated the PAM50 enrichment in the proteomic clusters in figure 2A and for the CoTCs, see figure S2C and S4J, respectively. For the 4 major clusters in figure 2A we see enrichment of PAM50 basal-like tumors in the proteomic basal-like cluster, Luminal A enrichment in the cluster with mainly luminal A and some HER2 and luminal B tumors, luminal B and HER2 enrichment in the combined luminal B and HER2 cluster and normal-like enrichment in the last proteome cluster. For the CoTC we see the same trend where each of the clusters are mainly enriched by one subtype except for CoTC6 with enrichment of both luminal B and HER2.

We acknowledge that it is sometimes difficult to define clusters based on the dendrogram. Therefore, protein clusters were defined using the correlation matrix (Fig S2B) and each cluster is indicated by vertical black bars in figure 2A and S2.

In all enrichment analysis performed we used the list of 9995 proteins identified and quantified across all tumors as background for enrichments. This has now been added to the materials and methods. Thanks for picking this up!

9. There are multiple gaps between novel peptides and their potential application as neoantigens. First, it is not clear whether they are expressed in normal tissues. Page 16, "CAGE and ribosomal profiling reveal they are transcribed and translated in other systems", what are the other systems? Were these peptides identified in any normal tissues? Secondly, MHC binding affinity of these peptides should be evaluated.

We agree that there is more information needed to use the novel identified peptides as neoantigens. However, in our view, a first important step is to identify potential neoantigens as translated protein products. The current workflows to identify neoantigens today do not rely on MS as a way of filtering for translated protein products that can be presented by MHC. Nonetheless, we are grateful for this question, since we believe that the added data and analysis as described below strengthen the overall message.

To investigate whether the novel peptides are expressed in normal tissues we concatenated the list of novel peptides identified in our data to Ensembl human protein database and searched the normal tissues of the draft proteome (Kim et al, 2014 Nature). We also

predicted MHC binding using MHCflurry (O'Donnell, 2018 Cell Systems). Of the 390 novel peptides, 117 (30%) remained as potential neoantigens after excluding novel peptides identified in the draft proteomes normal tissues and demanding peptides to be predicted to bind to MHC class I (Figure 7D).

To further investigate the tumor specificity of novel peptides, we expanded the analysis by adding an additional MS proteome dataset with 5 breast tumors with adjacent normal tissue. The data provides examples of novel peptides with increased levels in tumors compared to normal tissue and of increased levels in matched tumor-normal pairs. This suggests that there are both potential tumor specific and patient specific novel peptides (Fig S14H, I). An interesting observation in the tumor-normal pair data is that the lncRNA lnc-AKAP14-1:3 with high levels in one Luminal A tumor in the Oslo landscape cohort was identified with high levels in a luminal A tumor in the additional dataset.

The other systems referred to in the sentence "CAGE and ribosomal profiling reveal they are transcribed and translated in other systems" were detailed in the materials and methods and are THP-1 cell lines published by Fritsch et al 2012 Genome Research and mapped CAGE data across a panel of biological samples (975 human and 399 mouse samples, including primary cells, tissues and cancer cell lines) published by Forrest et al 2014 Nature. We have now added references also to the results section so it is easier to grasp what CAGE and ribosomal profiling data were used.

10. The authors tried very hard to justify that PAM50 genes have higher mRNA-protein correlations even after variance adjustment. However, a very complicated method was used for the adjustment and the method was not formally evaluated, and it remains unclear why PAM50 genes tend to have higher mRNA-protein correlation if it is not because of higher mRNA variation.

The selection of the best predictive mRNAs levels from large scale data sets that are good surrogate markers for protein level is an interesting finding, but potentially confounded by selection of tumors with big dynamic range of these mRNA and protein levels. Hence, we wanted to make sure we take this into consideration. However, we agree with the reviewer that our method might seem complicated and are very grateful for bringing this to our attention. We derived our model using an exploratory approach and this, together with the aim to make the results viewable in 2D, resulted in a perhaps overly complex model. We have now revised our analyses to adhere to a more standard statistics approach. We first realized that the saturation pattern we saw for mRNA-protein correlations (y) as a function $sd_{RNA}(x)$ and $sd_{protein}(z)$ is actually a sigmoidal (S-shaped) pattern (the lower portion of the 'S' is obscured by the low number of negative correlation values). This pattern is typical for response values limited by an interval, as indeed is the case here: the correlations are limited in the interval $[-1, 1]$. A standard way of modeling such response values (displaying a sigmoidal response pattern) is to use a generalized linear model (GLM) with a logit link function and a binomial distribution modeling the error term, i.e., the residuals ϵ (Lesaffre et al. 2007, Biostatistics). (We make a note of that this is the same model used for logistic regression, where the response displays an extreme variant of sigmoidal pattern only taking values either 0 or 1.) The logit function is defined on the $[0, 1]$ interval while our y is in the $[-1, 1]$ interval, but this is easily remedied by a simple rescaling, $y' = (y+1)/2$, and using y' in the GLM.

As before, the y -variable (correlations) appears to depend on both x (sd_{RNA}) and z ($sd_{protein}$) and we want to model this dependency. However, instead of creating a combined mRNA

and protein sd (sd_{comb}) as in the previous analysis, we now take the more straight-forward, and more standard, approach of creating a bivariate GLM: $logit(y')=\alpha+\beta x+\gamma z+\epsilon$. Analogous to our previous analysis, we then look at the residuals of this GLM and test for enrichment of very high (or low) correlation values for selected protein groups. We now feel that our analysis approach has become much clearer and adhere to standard statistics practice. Moreover, it turned out that the new approach also strengthened our results and provided a clearer picture for the biological conclusions. We are, therefore, again very grateful for the reviewer for raising this comment.

11. The median mRNA-proteomics correlation (Figure 5A, 0.45) and median RPPA-proteomics correlation (Figure S1J, 0.47) are similar. Why RPPA measurements were considered as consistent with MS-based measurements (page 4)? If so, is this also true for mRNA measurements?

We are grateful for the reviewer to notice this slip in the original sentence:

Both spectral search methods yield similar protein identifications (Fig S1I) whose quantities are consistent with RPPA findings (Fig S1J)

The new sentence which indicates the percentage of significant correlations now reads: Both spectral search methods yield similar protein identifications (Fig S1I), 60% of whose quantities are positively correlated with RPPA findings (Fig S1J)^{13,17}, and MS-based profiles of BC hallmark proteins are consistent with well established characteristics of tumor PAM50 classifications (Fig S1K).

Some of the reasons to the lack of correlation between MS and RPPA is mentioned in the figure text to Fig S1J.

12. Association between CoTC6 and glycolytic metabolites was used to support the added value of proteomics. This is interesting, but are any of the PAM50 subtypes associated with glycolytic metabolites? Did glycolytic enzymes/mRNAs show similar changes as the metabolites?

We claim the consistent grouping of these tumors by analyses of metabolite abundances and protein quantities illustrates the value added by proteome profiling because unbiased analysis of the proteome groups tumors in accordance with a direct measurement of phenotype (metabolite abundances), illustrating phenotypic differences can be inferred from the proteome.

We do not see an up-regulation of glycolytic enzymes at the protein or mRNA level in those tumors we designate as "glycolytic" based on analysis of tumor metabolite abundances (Fig R5). This is not surprising because flux through a metabolic pathways is often altered by regulating a different, but connected pathway. For example, flux through glycolysis could be enhanced by inhibition of the TCA cycle or a defect in oxidative phosphorylation, neither of which may be observable by protein abundances.

Figure R5: Volcano plots of protein and mRNA abundance ratios of “glycolytic” tumors to the collection of CoTC3, CoTC4, and CoTC5 designated tumors. Glycolytic and closely related proteins are represented as purple dots. $-\text{Log}_{10}(\text{p-values})$ are derived from two-tailed Students’ t-tests not corrected for multiple comparisons. No enrichment in the regulation of glycolytic proteins/transcripts is detected.

We hypothesize that the growth phenotype of these tumors (all PAM50 luminal B and HER2) creates a demand for glycolytic intermediates as precursors for biosynthetic reactions, and consumption of these precursors thermodynamically up-regulates flux through glycolysis, or that activation occurs at the post-translational level. If true, the proteins that are driving the growth or post-translational-modification changes in these tumors may be identifiable by examining the protein profiles. We are currently looking into this possibility, however, confidently identifying regulatory mechanisms requires extensive validation that is outside the scope of this breast cancer landscape study.

We understand that displaying an observation without follow-up may not be completely satisfying to a reader interested in breast-tumor metabolism. However our aim to display the utility of our multi-omics data-sets across fields of inquiry in breast cancer, which is achieved here because reviewer #2 is asking the exact sort of questions we hope to imotivate.

Reviewer 2 also inquires about the the association of PAM50 subtypes with the glycolytic phenotype. The group we designate as glycolytic are composed entirely of HER2 and Luminal B PAM50 subtype tumors. These designations are labeled as a color-bar underneath the column dendrogram of Figure 2F.

13. Figure 5C, the claim highly proliferative tumors had stronger mRNA-protein correlation needs to be tested statistically.

Please see combined responses to reviewer one's question:

12. For figure 5C, is the poor tumor-protein correlation at all linked to tumor purity levels?...

14. Proteomics data was quantified based on ratio to the tumor sample pool (equivalent to mean-centering). Was mRNA expression data also mean-centered before calculating tumor mRNA-protein correlation?

To clarify this question, we have added the following sentence to materials and methods section describing the RNA-protein correlation analysis:

The RNA data from the microarrays were log₂-transformed, quantile normalized and hospital-adjusted by subtracting from each probe value the mean probe value among samples from the same hospital as described by Aure et al¹⁶.

This information is also given at GEO where the RNA data used for analysis can be downloaded: <https://www.ncbi.nlm.nih.gov/geo/query/acc.cgi?acc=GSE80999>

15. Figure 3C, overlaying protein cluster characteristics from Fig2A to demonstrate functional coherence of the subnetworks over-estimates true functional association. Enrichment analysis should be performed based on real gene annotations and the data should be provided.

To confirm that the overlaid protein groups from fig A to the protein correlation network still harbor the same biological characteristics, we performed enrichment analysis as suggested. The data is provided in fig S7D, E and confirm the characteristics of the previously assigned protein groups for all but one group. The small transcription group (n=39) did not give any enrichment, but 9 proteins are GO annotated with transcription. The low number of proteins in the group makes it very difficult to reach statistical significance and the group is probably also influenced by luminal proteins since it associate closely to the luminal cluster in the protein correlation network and with clustering using all proteins.

16. The published metabolite analysis only quantified 18 metabolites?

Yes, that is correct. The benefit with high resolution magic angle spinning NMR spectroscopy (HR MAS) method to measure metabolites is that it keeps the tissue intact. However this comes with a limitation, as described by our co-author Haukaas in *Metabolites*, 2017 (<https://www.ncbi.nlm.nih.gov/pmc/articles/PMC5487989/>) "An important limitation of MR spectroscopy is its relatively low sensitivity (micromolar range compared to picomolar range for some MS-based methods), and thus the relatively few detectable metabolites..... Many of these are involved in pathways known to be important in cancer development and progression, predominantly glucose metabolism, amino acid metabolism and choline phospholipid metabolism, which are discussed in this review."

Since HR MAS can be used on intact tissue, the samples used for MS based proteomics are the same that were used for HR MAS, which is a benefit in terms of consistency in the analyses due to tumor heterogeneity.

17. Page 3, a reference is needed for the Oslo2 study

We see now that the references for the Oslo2 study was only mentioned in the materials and methods section in the sentence: The tumors from the Oslo2 cohort used in this study have previously been described ^{13,14}.

We have now added the references to page 3 and the sentence now reads:

Nine patients classified into each of the five PAM50 subtype groups were selected from the Oslo2 study cohort to ensure tumor diversity is represented (denoted Oslo2 Landscape cohort) (Fig 1)^{13,14}.

18. Page 6, the authors mentioned “The luminal tumor dominated CoTC groups (3, 4, and 6)”. However, the CoTC group 6 seems to be dominated by Her2 rather than luminal.

Thanks for spotting this miss. The sentence now reads:

The luminal and HER2 dominated CoTC groups (3, 4, and 6) are stratified by differential enrichment for proteins related to the estrogen response, E2F targets, G2M checkpoint proteins, and MYC targets (Fig 2F).

Reviewer #3:

Major revisions (arbitrary order):

In the proteogenomic analysis section of the manuscript and in the methods the authors describe the use of the human genome version GRCh37 and gene annotation GENCODE v19 with both being outdated versions since end of 2013. While the authors validate their potential novel findings with synthetic peptides, advancements in genome annotation are neglected (this includes the transition to GRCh38 and additional reassignment of non-coding genes to protein coding as highlighted in Wright et al., Nature Communications (2016) using the draft human proteome maps from Kim et al., Nature (2014) and Wilhelm et al., Nature (2014)). A quick scan of the claimed novel peptides revealed 10% of those now map to protein coding genes in Ensembl release 92 and 1% with partially deleted genomic sequence in the novel genome assembly.

The annotation of some of the novel findings is in part due to the time it takes to analyse and assemble all the different data types to a multi-omics landscape study. We also used the older genome version GRCh37 since some of the databases had not updated their annotation to the latest genome version when the proteogenomics searches were performed. Upon checking these databases they now include the latest human genome build and we performed the proteogenomics analysis with the updated databases (Uniprot+Ensembl 92+GENCODE 28, updates in 2018), which revealed that 30 of 390 peptides are now part of protein coding gene annotations. All the peptides identified in the old version are also identified in the new. The identification of novel peptides from previously un-annotated, but now annotated genomic regions function as a validation of the method's ability to find novel protein coding regions. The shift in status from novel to annotated during preparation of the manuscript have been noted in the supplementary table as well as the new genome coordinates for GRCh38.

Meanwhile, we have updated our pipeline to latest genome assembly and GENCODE database (<https://github.com/lehtiolab/proteogenomics-analysis-workflow>).

Additional 32% of novel peptides map to known sequences when accounting for an amino acid substitution.

The authors should incorporate an additional step of filtering for novel peptide calls and check if the regions have been included in recent updates to the annotation as well as for additional SNPs in the cancer samples.

We intentionally kept the novel peptides with single mismatches to known proteins. In contrast to other proteogenomics studies, as described in materials and methods, we performed two additional analysis steps to strengthen the confidence in single-substitution novel peptides. First, SpectrumAI (please see Zhu Y. et al Nature communications 2018 for detailed evaluation of the method) was used to validate the presence of flanking MS2 peaks in the experimental spectra to support the substituted amino acid. Second, novel peptides were screened against a peptide database derived from nsSNP to make sure novel peptides are not results of missense SNPs.

Regarding the proteogenomic analysis: the authors use separate databases for their searches. This can lead to identification of different peptides for the same spectrum. The authors should incorporate a validation step and check overlap of spectra between their

novel, SAAV, and reference peptide identifications and attribute these spectra to the already identified sequences.

The two parallel searches, pI-restricted 6FT and VarDB search, used only one database that included a known human protein database from Ensembl to allow novel and SAAV peptides compete with known peptides in spectra matching process. In the subsequent calculation of class FDR, we removed identifications from known peptides and kept only novel and SAAV. Please see Zhu Y. et al Nature communications 2018 for detailed evaluation of the method.

The authors are rather stingy when it comes to citing other papers. For instance, several important papers on proteogenomics (incl. some published in Nature Communications) are missing. Similarly, proper references for other methods such as FASP or RPPA missing. We are aware that all contributions in the multiple of fields covered in this paper is not cited. Key contributions have been selected as far as possible, but there will always be a limitation and thus compromise to the number of references cited due to length restrictions. Without knowing exactly the papers in Nature Communication the reviewer refers to, we assume it is Huang et al 2017 and Wright 2016. Huang et al use a PDX model and was not cited since we use tumors directly from patients and in their study they did not have the luminal A and normal-like subtypes. Wright describes a proteogenomics method. Our proteogenomics method is described in Zhu et al 2018 Nature Communications, in which we cite relevant proteogenomics studies. Zhu et al considers the stringent criteria described in Wright et al and sets additional criteria by spectral annotation to improve SNP and mutation detection on protein level.

Regarding FASP, we use a modified version of FASP described in the now clearly cited paper Branca et al 2014. We agree with the reviewer that the original paper describing filter aided sample preparation should be cited, and thus Manza Proteomics 2005 is now cited in the paper.

The RPPA data used in this paper was not generated solely for this paper but still represent the same tumors analysed with MS proteomics. RPPA data for the tumors analysed by MS proteomics from the Oslo2 cohort were from the cited paper Aure et al 2017 Breast Cancer Res, in which details about RPPA data generation can be found.

The introduction of the PAM50 panel misses the important aspect that it resulted in 5 different classification. The number '5' only suddenly shows up when the sample cohort is described.

Good point; the 5 subtypes have now been described in the introduction.

In the results section for protein correlation analysis (page 5 and 10) the authors use subjective terminology such as clearly and robustly. Looking at the figures I do not agree with this assessment and would encourage the authors to justify their insights in a quantitative manner, e.g. using similarity measures and proper statistical evaluations between clusters. Similarly, the authors seem to miss to mention that the co-elevation of EGFR and MET appears to be confined to ductal carcinoma in situ regions FOR A SUBSET of normal like tumors. Figure 4E clearly shows that this is the case for 3 cases, but not for the 6 other cases. Please be more concise and evaluate the statistical significance of this observation.

This is a relevant point in regards to scientific integrity of presented work. We apologize for using subjective terminology as clearly and robustly. We have now read through the text and removed them.

Statistical evaluation between clusters has now been added and is presented in Figure S2C and S4J. For the 4 major clusters in figure 2A we see enrichment of PAM50 basal-like tumors in the proteomic basal-like cluster, Luminal A enrichment in the cluster with mainly luminal A and some HER2 and luminal B tumors, luminal B and HER2 enrichment in the combined luminal B and HER2 cluster and normal-like enrichment in the last proteome cluster. For the CoTC we see the same trend where each of the clusters are mainly enriched by one subtype except for CoTC6 with enrichment of both luminal B and HER2.

We have now re-evaluated the IHC scoring presented in Fig 4E, F, which confirmed the previous scoring. There is indeed a subset of normal-like tumors which co-stain for EGFR and MET using antibodies. The IHC data from our 2 cohorts suggest a previously unknown heterogeneity in the normal-like subtype, which shed some light on this controversial subtype, and warrant further investigations.

To clarify the IHC data, the result part for co-staining of EGFR and MET in normal-like tumors now reads:

“co-elevation of EGFR and MET appears to be confined to ductal carcinoma in situ (DCIS) regions for a subset of normal-like tumors”

Figure 5B: Since the different tumor types are spread over the entire range of proteome/transcriptome correlations, I don't think the authors' claim on page 12/13 (last/first paragraph) is justified. For instance, basal like and HER2 BC types are spread over the entire range. Please, tone down this statement and/or provide statistical support for this claim.

Please see combined responses to reviewer one's question:

12. For figure 5C, is the poor tumor-protein correlation at all linked to tumor purity levels?...

Figure 2: In the CoTC analysis, it seems that only 29 samples were considered. It is, however, not clear which 29 samples were used and which 16 samples were dropped and for which reason. Please explain.

Please see the combined answer with reviewer one's question:

8. Is there concern with the limited sample numbers after reducing to the subset of samples, that the clustering is more driven by individual differences.

Figure 2F: It is not clear which branch corresponds to which CoTC number – please provide this information.

We are puzzled about this question, since 2F refers to the enrichment figure. The previous figures (C, D) with branches are color coded for CoTC (C) and PAM50 (D). The color codes can be found on the top right side of the panel figure.

The manuscript describes a true 'tour de force' which generated HUGE amounts of data. It seems that the authors try to show as much as possible in the figures. The result, unfortunately, is that many panels and their labeling is far too small to be useful. Thus, I

strongly recommend to delete some of the panels and to enlarge the fonts to a legible size. Thanks for the comment! We perceive it as a tradeoff between showing enough information to make a convincing message and simplifying the data presentation. During review, the manuscript has been reworked, and figures have been remade and removed and fonts enlarged. For example, fig 2E is now S5A, fig 5C is reworked and simplified and is now 5D. However, based on review questions, we have also changed and added data and figures, but additions mainly to the supplementaries. Also, all images will be submitted in vector format to enable formatting by Nature communications (e.g. if font is perceived to small) if necessary.

Minor revisions/typos (arbitrary order):

The authors alternate between terminologies using gene and proteins almost synonymously between the main manuscript and the supplementary figures. This leads to confusions specifically when it comes to the numbers of proteins/genes identified. Examples here is the identification of 13,997 protein products and 12,645 genes in the results section but the use of the term 'protein identifications' in Fig. S2 I when in fact 12645 genes are shown in the figure.

Thanks for pointing this out. In figure S2I we have now clarified that we compare proteins based on gene symbols.

In the materials and methods section we have a definition about how we performed protein quantification and defined protein:

Protein quantification by TMT10plex reporter ions was calculated using TMT PSM ratios to the tumor sample pool and each tumor were normalized to its median ratio. The median PSM TMT reporter ratio from peptides unique to a gene symbol was used for quantification. Protein ratios to pool of samples is denoted as protein abundance or protein levels in figures and text. For subsequent bioinformatic analysis we used a subset of 9995 gene symbols, representing proteins, with TMT quantification across all sets.

The previous sentence in the results section:

The subset of proteins from 9,995 genes that were quantified in each of the 45 tumors (i.e., the quantified proteome) is used for all quantitative proteome analyses (Fig S1C-H).

Have now been changed to clarify our definition of protein in the paper:

The subset of 9995 proteins quantified in each of the 45 tumors, based on gene symbol centric quantification (denoted proteins henceforth), is used for all quantitative proteome analyses (i.e., the quantified proteome)(Fig S1C-H).

Abstract, line 4, reads proteome-based networks distinguishes BC subtype-specific functional protein modules; should be proteome-based networks distinguish

Thanks for the correction!

Page 16, paragraph 3; OSLO2 should be consistent with the remaining manuscript and read Oslo2

Thanks, we have now changed the text to Oslo2.

Figure 7 D: the three marked peptides belong to Inc-AKAP14-1:1 and not Inc-AKAP14-1:3
Thanks for pointing out this error, indeed the supplementary table S4 shows they belong to Inc-AKAP14-1:1. After further examination of the sequences of both lncRNAs, the three peptides are actually shared between Inc-AKAP14-1:1 (ORF2) and Inc-AKAP14-1:3 (ORF1). Because there is a fourth peptide unique to Inc-AKAP14-1:3 (ORF1), we think it is more appropriate to still use Inc-AKAP14-1:3 to represent them based on Occam's Razor rule. We have fixed this in supplementary table S4 as well.

Page 5, reference to figure S2 (tumor composition correlated between MS and histopathology based evaluation (Fig S2G-I), should properly be Fig S2H-J.
Thanks, this has now been updated. In the reworked version of the paper, the figures are now S3H-J.

Figure 4B: legend for the color codes is missing.
Thanks, true, we have now added a bar to indicate how much each gene correlate to another.

Figure 5B: 2 of the p-values are shifted to the right.
The p-values in figure 5B are in the correct places. Oncotype Dx did not reach significance even if median is much higher than the median for all mRNA-protein correlation. Possibly due to that some genes in the panel have low correlation and the number of genes in the panel is quite low (n=21, compared to 50 and 70 for PAM50 and MammaPrint, respectively).

Reviewers' Comments:

Reviewer #1:

Remarks to the Author:

The authors addressed most of my comments. I have a few comments.

Updated figure 4D is much more informative. Thank you. But for 4E and F, I still think there maybe a way to make it easier to view. In 4E and 4F, the goal is to show the increase of concordance of MET and EGCR in the DCIS of normal-like. For the main figure, is there a need to show the other subtypes and to show the samples without DCIS component? Of the normal-like this there are only 5 that were listed as also having some % of DCIS component. And in 5F only 6 normal-like seem to have DCIS. I don't think you need the other tumor types since that isn't your point and the updated figure 4D shows the co-expression is predominantly in the basal and normal-like.

I'm not sure if we were discussing the same thing for the Figure 5B response. I was asking if it was related to how the genes were quantified as highly similar genes are often difficult to map and smaller genes are often missed in sequencing depending on the fragment size of the library. I typically see more spliceosome genes in FFPE samples where the inherent fragment size is smaller than what we typically fragment the Fresh Frozen RNAs too. This was more of a thought question that I was wondering if the difference was based on the difficulty of accurately quantifying the genes in the RNA, not about the variability.

minor

1. S5E – there appears to be a mistype, I believe you mean CoTC4 instead of CoTC2.

Reviewer #2:

Remarks to the Author:

The authors have addressed most of my questions. However, I still have concerns about the clustering/subtyping section, especially the identification of 6 subtypes using 29 samples. Again, the sample size is just too small to allow useful subtype analysis. The authors tried to justify the small sample size using high proteomic depth. While I applaud the analytical depth achieved by this study, it does not help with the sample size problem. Actually, increased number of variables will require increased number of samples for robust subtype detection. This problem could be potentially alleviated if the subtypes are reproducible in an independent cohort, e.g. the CPTAC cohort. However, as explained by the authors, validation cannot be performed due to technical difficulties. Thus, we are left with a new breast cancer subtype classification system with unknown robustness and generalizability. It is even unclear what is the proportion of breast tumors that can be represented by these 6 subtypes, as 36% of the samples (16/45) were considered as "outliers". In summary, it is difficult to imagine how this can benefit future breast cancer studies. My suggestion is to either find a way to validate this subtype classification or drop it from the manuscript.

Another question is about the new analysis on MHC binding affinity prediction. It is not clear whether HLA typing was performed. This should be described in the method section.

Reviewer #3:

Remarks to the Author:

We appreciate the time and effort of the authors to revise their manuscript and think that almost all issues raised by our initial review were appropriately addressed. 'Almost all' as there are two issues that require some clarification:

1) Reviewer #3, 2. Comment:

The authors highlight in their reply that they used SpectrumAI to validate the nsSNP PSMs and also screened novel peptides (from pseudogenes, lncRNAs etc) against a peptide database derived from nsSNP to make sure novel peptides are not results of missense SNPs. However, the second step seems to be missing in the manuscript and/or method section. Please clarify.

2) Reviewer #3, 3. Comment:

In the initial review, we were referring to different peptide sequence assignments for the same spectrum through the use of two separate databases. In the reply the authors state that the "... two parallel searches [...] used only one database that included a known human protein database from Ensembl to allow novel and SAAV peptides compete with known peptides... ". The authors also refer to their proteogenomics paper for further details without any changes to the methods section. However, when looking at the cited paper, the proteogenomics search section is almost identical and thus does not shine any new light on the subject. Please clarify this issue as well.

Reviewers' comments:

Reviewer #1:

The authors addressed most of my comments. I have a few comments.

1. Updated figure 4D is much more informative. Thank you. But for 4E and F, I still think there maybe a way to make it easier to view. In 4E and 4F, the goal is to show the increase of concordance of MET and EGCR in the DCIS of normal-like. For the main figure, is there a need to show the other subtypes and to show the samples without DCIS component? Of the normal-like this there are only 5 that were listed as also having some % of DCIS component. And in 5F only 6 normal-like seem to have DCIS. I don't think you need the other tumor types since that isn't your point and the updated figure 4D shows the co-expression is predominantly in the basal and normal-like.

We are grateful for this suggestion. However, we think the value of using coexpression of EGFR and MET as a marker for the *in situ* component of normal-like tumors is more apparent when it is contrasted with the staining pattern of all subtypes. We are aware that the image is a bit complicated, but it shows all the data in a transparent way without the need to consult the supplementary figures.

2. I'm not sure if we were discussing the same thing for the Figure 5B response. I was asking if it was related to how the genes were quantified as highly similar genes are often difficult to map and smaller genes are often missed in sequencing depending on the fragment size of the library. I typically see more spliceosome genes in FFPE samples where the inherent fragment size is smaller than what we typically fragment the Fresh Frozen RNAs too. This was more of a thought question that I was wondering if the difference was based on the difficulty of accurately quantifying the genes in the RNA, not about the variability.

We appreciate the reviewer's clarifying comment on the question and concern on the lower mRNA-protein correlation groups of genes. For the samples in question we have mRNA expression data from fresh frozen tissue, not paraffin-embedded, so fragment size is not an issue here as it could be if it was data from FFPE tissue. Also, there is no mRNA fragmentation step in the Agilent gene expression array workflow and therefore, when working with intact RNA samples, fragment size shouldn't be an issue.

We do acknowledge the reviewer's point about that the sequence similarity of the ribosomal genes may contribute and explain some of the lower correlation in this group. However, since we are using microarray data and not RNA-seq we do not have the same challenges related to mapping. The 60-mer probes that Agilent have designed should be optimized to promote specificity and be able to distinguish between the ribosomal genes. The microarray workflow uses a linear in-vitro transcription amplification which is less error prone than multiplicative PCR amplification and there are fewer steps from total RNA to Cy-labeled cRNA than from total RNA

to cDNA RNA-Seq libraries which also reduces the chance of errors or bias to occur.

Minor

1. S5E – there appears to be a mistype, I believe you mean CoTC4 instead of CoTC2.

Thank you for spotting this! We have changed it.

Reviewer #2:

1. The authors have addressed most of my questions. However, I still have concerns about the clustering/subtyping section, especially the identification of 6 subtypes using 29 samples. Again, the sample size is just too small to allow useful subtype analysis. The authors tried to justify the small sample size using high proteomic depth. While I applaud the analytical depth achieved by this study, it does not help with the sample size problem. Actually, increased number of variables will require increased number of samples for robust subtype detection. This problem could be potentially alleviated if the subtypes are reproducible in an independent cohort, e.g. the CPTAC cohort. However, as explained by the authors, validation cannot be performed due to technical difficulties. Thus, we are left with a new breast cancer subtype classification system with unknown robustness and generalizability. It is even unclear what is the proportion of breast tumors that can be represented by these 6 subtypes, as 36% of the samples (16/45) were considered as “outliers”. In summary, it is difficult to imagine how this can benefit future breast cancer studies. My suggestion is to either find a way to validate this subtype classification or drop it from the manuscript.

We acknowledge the reviewers point about subtype classification. With the current data and lack of means of proteome based validation we can not propose a new subtype classification. Our intention with the analysis is to provide an unbiased analysis of proteome based tumor grouping in the Oslo2 landscape cohort. To clarify this we have changed the text so that the words tumor grouping/clustering is used instead of subtype for the proteome data.

The following sentences have been changed:

In the abstract:

Proteome-based networks distinguish BC subtype-specific functional protein modules, with co-expression of EGFR and MET marking ductal carcinoma in situ regions of normal-like tumors and lending to a more accurate classification of this poorly defined subtype.

To:

Proteome-based networks distinguish functional protein modules for breast tumor groups, with co-expression of EGFR and MET marking ductal carcinoma in situ regions of normal-like tumors and lending to a more accurate classification of this poorly defined subtype.

In the result section describing the CoTC:

Core sets of tumors whose proteomes are representative of a proteome-based subtype assignment are defined to address this ambiguity, allowing all tumor proteomes to be described in relation to their core-group composition. Unsupervised clustering based on high variance protein (n=1334) abundance profiles (Supplemental Methods) produces six consensus core tumor clusters (CoTC) (Fig 2C, S4C-K).

To:

Core sets of tumors whose proteomes are representative of a proteome-based grouping are defined using unsupervised clustering based on high variance protein (n=1334) abundance profiles (Supplemental Methods), which produces six consensus core tumor clusters (CoTC) (Fig 2C, S4C-K).

However, it's fair from the reviewer's side to question the number of cases related to these proteome defined groups. To elucidate this, we selected high RNA-protein correlating genes, used for protein subgrouping, as surrogate markers to study the grouping in a large breast cancer cohort (TCGA, Nature 2012). Clustering of 1100 tumors with RNA-seq data from TCGA using these genes corresponding to the high variance proteins used for generation of the CoTC groups generated 4 major groups (Fig R1). These groups resemble the major CoTC groups (in parenthesis) with a basal-like group (CoTC1), a luminal A group (CoTC3), a normal-like group (CoTC5) and a mixed luminal B-HER2 group (CoTC6). There is presence of an immune RNA cluster mainly in basal-like and mixed luminal B-HER2 on number of tumor samples. However, the immune RNA cluster influence is not strong enough on RNA level to group the immune-rich tumors as observed in the protein level data (for CoTC2 and CoTC4). In figure S5 we show also the presence of tumors with immune characteristics of CoTC2 and CoTC4 in the Oslo2 cohort.

Immune infiltration is not considered by PAM50 classification. However, immune infiltration is known to be linked to outcome in breast cancer (see for example Denkert 2018 Lancet Oncology). We believe that the proteome based observations of immune related tumor groups adds valuable knowledge to the discussion of immune infiltration in breast cancer.

In conclusion, we agree with the reviewer, and have altered the manuscript so that we do not refer these cases as tumor subtypes, but present the protein driven clustering, and have gathered supporting evidence that immune proteome defines cases which can be found in numbers in a larger cohort.

Figure R1. Clustering of TCGA RNA-seq data using genes overlapping with high variance proteins used to define CoTC. The RNAs for clustering were selected using the following criteria, 1) identified and quantified among the high variance proteins used to define CoTC, 2) significant positive mRNA-protein correlation (Bonferroni adjusted p-value <0.05). This filtering resulted in 923 RNAs from the original 1334 proteins (all gene symbol based).

Another question is about the new analysis on MHC binding affinity prediction. It is not clear whether HLA typing was performed. This should be described in the method section.

We have clarified this point in the m&m now. The HLA typing is not done for these patients. Hence, all the potential neoantigens may not bind a HLA type present in patients' tumor. On the other hand, different HLA type peptide binding characteristics are only beginning to be unravelled. See for example Abelin et al 2017 Immunity for a good approach to start to understand the characteristics underlying peptide binding to each HLA type.

We have now added the following description to m&m under the section; Search neoantigen candidates in draft proteome data and MHC binding prediction:

No HLA typing was performed. The HLA supertype representatives (A01:0, A02:01, A03:01, A24:02, A26:01, B07:02, B08:01, B27:05, B39:01, B40:01, B58:01, B15:01) were used to predict affinity.

Reviewer #3:

We appreciate the time and effort of the authors to revise their manuscript and think that almost all issues raised by our initial review were appropriately addressed. 'Almost all' as there are two issues that require some clarification:

1) Reviewer #3, 2. Comment:

The authors highlight in their reply that they used SpectrumAI to validate the nsSNP PSMs and

also screened novel peptides (from pseudogenes, lncRNAs etc) against a peptide database derived from nsSNP to make sure novel peptides are not results of missense SNPs. However, the second step seems to be missing in the manuscript and/or method section. Please clarify.

Thanks for pointing this out. We have now made this clearer by adding the following description to the materials and method section under Proteogenomics search and Class-specific FDR: Novel peptides matching to sequences from nsSNPs in CanProVar 2.0 were marked in supplementary table S6. None of the peptides that could be explained by CanProVar were found in the samples by iCogs SNP array data.

2) Reviewer #3, 3. Comment:

In the initial review, we were referring to different peptide sequence assignments for the same spectrum through the use of two separate databases. In the reply the authors state that the "... two parallel searches [...] used only one database that included a known human protein database from Ensembl to allow novel and SAAV peptides compete with known peptides...". The authors also refer to their proteogenomics paper for further details without any changes to the methods section. However, when looking at the cited paper, the proteogenomics search section is almost identical and thus does not shine any new light on the subject. Please clarify this issue as well.

We are grateful for the reviewer's diligence in this question. Upon checking the scan numbers for peptides matched in either the VarDb or 6FT search, we identify 1 spectrum with duplicate explanations. These 2 novel peptide cases have now been removed and supplementary table and figures updated.

Reviewers' Comments:

Reviewer #1:

Remarks to the Author:

my questions have been addressed

Reviewer #2:

Remarks to the Author:

It should be more meaningful to perform HLA typing for individual patients and use that information to guide MHC binding prediction. However, as this is not a critical part of the study, and the fact that HLA typing was not performed is now described in the methods section, I am ok with it.